# Purkinje cell intrinsic activity shapes cerebellar development and function

Catarina Osório [1], Joshua J. White[1], Paula Torrents-Solé [1], Jie Yang[1], Nienke Mandemaker[1], Federico Olivero[1], Freya Kirwan [1], Laura Post[1], Zahra Hemmat[1], Fred de Winter [2], Eleonora Regolo [1], Francesca Romana Fiocchi[1], Inês Serra[1], Saffira Tjon[1], Zeliha Ozgur[3], Mirjam C.G.N. van den Hout [3], Wilfred F. J. van IJcken [3], Guillermina López-Bendito [4], Aleksandra Badura [1], Lynette Lim[5,6], Geeske M. van Woerden [1,7,8] & Martijn Schonewille [1]

The emergence of functional cerebellar circuits is heavily influenced by activity-dependent processes. However, the contribution of intrinsic Purkinje cell activity to cerebellar development remains less understood. Here, we demonstrate that before synaptic networks mature, Purkinje cell intrinsic activity is essential for regulating dendritic growth, establishing connections with cerebellar nuclei, and ensuring proper cerebellar function. Disrupting this activity during the postnatal period impairs motor function, with earlier perturbations causing more severe deficits. Importantly, only early developmental disruptions lead to pronounced defects in cellular morphology, highlighting key temporal windows for dendritic growth and maturation. Transcriptomic analyses reveal that early intrinsic activity drives the expression of activity-dependent genes, including *Prkcg* and *Car8*, which are essential for dendritic development. Our findings emphasize the importance of temporally regulated intrinsic activity in Purkinje cells in guiding cerebellar circuit development, providing a potential unifying mechanism underlying cerebellum-associated disorders.

Neural development relies on a complex interplay between genetic and early activity-dependent processes that guide morphogenesis and establish functional neural circuits[1,2]. Early electrical activity regulates a wide range of developmental programs in various regions of the nervous system, including the retina[3,4], spinal cord[5,6], cochlea[7], and neocortex[8,9]. Genetic programs are especially relevant during cerebellar ontogenesis, where more than half of the brain's neurons[10] converge to form an intricate network of connections.

The cerebellum undergoes a prolonged maturation period, making it particularly susceptible to developmental errors[11]. Indeed, evidence suggests that a common feature of several cerebellar-associated disorders is the impairment of Purkinje cell intrinsic activity[12], which is the spontaneous activity generated by these principal neurons in the

[1]Department of Neuroscience, Erasmus Medical Center, Rotterdam, The Netherlands. [2]Laboratory for Regeneration of Sensorimotor Systems, Netherlands Institute for Neuroscience, Royal Netherlands Academy of Arts and Sciences (KNAW), Amsterdam, The Netherlands. [3]Center for Biomics, Erasmus University Medical Center, Rotterdam, The Netherlands. [4]Instituto de Neurociencias de Alicante, Universidad Miguel Hernández-Consejo Superior de Investigaciones Científicas (UMH-CSIC), Sant Joan d'Alacant, Spain. [5]VIB Center for Brain and Disease, Leuven, Belgium. [6]Department of Neurosciences, Katholieke Universiteit (KU) Leuven, Leuven, Belgium. [7]Department of Clinical Genetics, Erasmus Medical Center, Rotterdam, The Netherlands. [8]The ENCORE Expertise Center for Neurodevelopmental Disorders, Erasmus Medical Center, Rotterdam, The Netherlands. ✉e-mail: c.osorio@erasmusmc.nl; m.schonewille@erasmusmc.nl

absence of synaptic inputs[13,14]. However, the requirement for early Purkinje cell activity in their maturation and its potential contribution to cerebellar dysfunction remains largely unexplored. Purkinje cells are at the core of cerebellar circuitry, and their development is intertwined with that of the surrounding cerebellar elements[15]. The genetic and cellular programs governing the genesis, migration, diversity, and differentiation of cerebellar cells have been explored and cataloged through recent single-cell transcriptomic endeavors[16,17]. Although less studied, activity-dependent mechanisms also play a crucial role in the early stages of cerebellar development. For example, the organization of Purkinje cell parasagittal banding requires Purkinje cell output signals[18], while the competition and subsequent elimination of climbing fiber afferents depend on Purkinje cell activity[19]. Furthermore, electrical stimulation of cerebellar slices induces the expression of *Arc* in Purkinje cells, contributing to the elimination of climbing fiber synapses[20]. Recent in vitro studies have revealed that even when synaptic transmission is blocked, Purkinje cells exhibit intrinsic activity as early as postnatal day 3, with this activity gradually increasing until it stabilizes at mature levels by the end of the second postnatal week[21]. Nevertheless, the extent to which early intrinsic activity influences the molecular programs underlying Purkinje cell maturation, circuit assembly, and overall cerebellar function remains unclear.

To investigate the role of Purkinje cell intrinsic activity in cerebellar development, we genetically reduced their intrinsic activity at distinct postnatal stages in mice. Our findings reveal that early activity is fundamental for the development of Purkinje cell dendritic arbors, axonal targeting of cerebellar nuclei neurons, and the acquisition of cerebellar functions. Intrinsic activity is particularly important during the first two postnatal weeks for Purkinje cell maturation and the establishment of balance and coordination functions. Molecularly, early intrinsic activity modulates the expression of several activity-dependent genes, including *Prkcg* and *Car8*. Functional shRNA knockdown of *Prkcg* and *Car8* genes demonstrates that they are key regulators of Purkinje cell development. Overall, our results unveiled a selective requirement for early Purkinje cell activity in shaping cerebellar development and function.

## Results

### Overexpression of Kir2.1 reduces Purkinje cell intrinsic activity
To reduce the intrinsic activity of Purkinje cells during postnatal development, we generated a mouse model that allows for temporal control of activity. The tamoxifen-inducible *Pcp2^creER* mouse line[22] was crossed with a conditional line expressing the inward rectifying potassium channel 2.1 (*Kir2.1*) fused to the *mCherry* reporter gene[23], generating *Pcp2^creER*;*Kir2.1* mutant mice (Supplementary Fig. 1a, c). As an inducible control, *Pcp2^creER*;*Ai14*[24] mice conditionally expressing the *tdTomato* reporter were also generated (Supplementary Fig. 1a, b). To examine Purkinje cell activity in a cell-autonomous manner, we administered a single low dose of tamoxifen at postnatal day (P) 1 to achieve sparse labeling of Purkinje cells expressing either Kir2.1-mCherry or tdTomato in mutant and control pups, respectively (Fig. 1a and Supplementary Fig. 1a, d). At P21, when cerebellar development is complete, we performed ex vivo recordings from sagittal cerebellar slices to evaluate the levels of intrinsic activity in Purkinje cells expressing

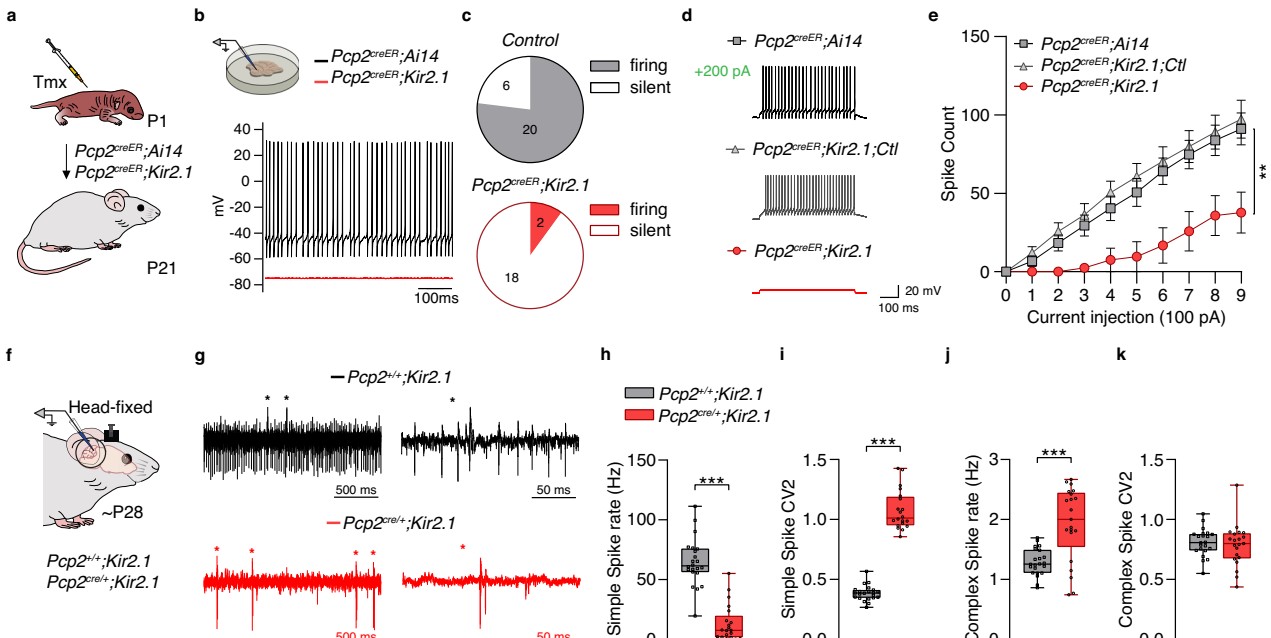

**Fig. 1 | Overexpression of Kir2.1 reduces the intrinsic activity of Purkinje cells at P21. a** Experimental design. *Pcp2^creER*;*Ai14* and *Pcp2^creER*;*Kir2.1* pups were injected with tamoxifen (Tmx) at postnatal day 1 (P1) to selectively express tdTomato or Kir2.1-mCherry in Purkinje cells, respectively. **b** Brains were collected at P21 for slice electrophysiology. Example traces of action potentials recorded from *Pcp2^creER*;*Ai14* (black) and *Pcp2^creER*;*Kir2.1* (red) Purkinje cells in the presence of NMDA receptor antagonist D-AP5, AMPA receptor antagonist NBQX, and GABA_A/glycine receptor antagonist picrotoxin. **c** Number of firing and silent Purkinje cells in *Pcp2^creER*;*Ai14* and *Pcp2^creER*;*Kir2.1* mice at P21. **d** Representative voltage responses to 200 pA step current injections from *Pcp2^creER*;*Ai14*, *Pcp2^creER*;*Kir2.1,Ctl*, and *Pcp2^creER*;*Kir2.1* Purkinje cells at P21. **e** Spike count of Purkinje cells from *Pcp2^creER*;*Ai14* (n = 13 cells/5 mice), *Pcp2^creER*;*Kir2.1,Ctl* (n = 10 cells/4 mice) and *Pcp2^creER*;*Kir2.1* (n = 9 cells/5 mice) at P21 in response to 100 pA incremental current injections. Two-way repeated measures ANOVA: $^{**}P < 0.01$. **f** Experimental design for in vivo recordings. Extracellular recordings were made from Purkinje cells in awake P28 ± 3 days *Pcp2^{+/+}*;*Kir2.1* (black) and *Pcp2^{cre/+}*;*Kir2.1* (red) mice. **g** Example traces of Purkinje cells recorded in *Pcp2^{+/+}*;*Kir2.1* and *Pcp2^{cre/+}*;*Kir2.1* mice. Complex spikes are identified with an asterisk. **h** Simple spike firing rate, **i** coefficient of variation 2 (CV2) for simple spikes, **j** complex spike firing rate, and **k** CV2 for complex spikes in Purkinje cells from *Pcp2^{+/+}*;*Kir2.1* (n = 22 cells/5 mice) and *Pcp2^{cre/+}*;*Kir2.1* mice (n = 21 cells/5 mice). Mann–Whitney U-test and unpaired Student's t-test with Welch's correction: $^{***}P < 0.001$. Box plots indicate the median (middle line), 25th and 75th percentiles (box), and 5th and 95th percentiles (whiskers) (**h**–**k**). Data are shown as the mean ± s.e.m. (**e**). Statistical details are provided in Supplementary Table 1. Source data are provided as a Source data file. Ctl control.

tdTomato (*Pcp2^{creER};Ai14*), Kir2.1-mCherry (*Pcp2^{creER};Kir2.1*), or neither (*Pcp2^{creER};Kir2.1;Ctl*, unlabeled controls). Recordings were conducted in the presence of synaptic blockers for NMDA, AMPA, GABA$_A$, and glycine receptors. In control recordings, 77% of Purkinje cells exhibited spontaneous firing, confirming intrinsic activity. In contrast, Kir2.1 overexpression abolished spontaneous firing in 90% of the recorded *Pcp2^{creER};Kir2.1* Purkinje cells by hyperpolarizing the resting membrane potential (Fig. 1b, c). Treatment with 300 μM barium, a potassium channel blocker, restored the membrane potential to control levels in *Pcp2^{creER};Kir2.1* Purkinje cells (Supplementary Fig. 2a). Kir2.1 overexpression also caused a significant shift in the current–voltage (I–V) curve of mutant Purkinje cells compared to controls, notably more hyperpolarized current at −140 mV. This I–V curve could also be restored to control levels by the presence of barium (Supplementary Fig. 2b–d). Despite the loss of intrinsic activity, mutant *Pcp2^{creER};Kir2.1* Purkinje cells retained the ability to fire action potentials in response to depolarizing current injection (Fig. 1d, e). However, their lower input resistance (Supplementary Fig. 2e) made them less excitable, requiring larger current injections (≥600 pA) to evoke the firing of action potentials (Fig. 1e and Supplementary Fig. 2f).

Next, we examined Purkinje cell activity in vivo using *Pcp2^{cre/+};Kir2.1* mice, in which Kir2.1 is overexpressed in all Purkinje cells, and control littermates *Pcp2^{+/+};Kir2.1* at P28 ± 3 days (Fig. 1f and Supplementary Fig. 1e–g). Purkinje cells were identified by the presence of simple and complex spikes, and single unit recordings were confirmed by the presence of the characteristic pause in simple spikes following each complex spike (Fig. 1g). The simple spike firing rate was significantly reduced in *Pcp2^{cre/+};Kir2.1* Purkinje cells (65 ± 4 Hz) compared with controls (13 ± 3 Hz) (Fig. 1g, h), and mutant cells displayed increased irregularity in simple spike firing (Fig. 1i). Conversely, the complex spike rate was significantly higher in *Pcp2^{cre/+};Kir2.1* Purkinje cells than in controls (Fig. 1j), while their firing regularity remained unchanged (Fig. 1k). In line with previous findings that simple spike firing rate primarily reflects the intrinsic activity of Purkinje cells[25], these results demonstrated that Kir2.1-overexpression also minimizes Purkinje cell firing activity in vivo. Taken together, these findings indicate that Purkinje cells overexpressing Kir2.1 exhibit reduced intrinsic activity but retain the capacity to generate action potentials when stimulated.

## Loss of intrinsic activity impairs Purkinje cell morphological maturation

To investigate whether disrupting intrinsic activity impacts the morphological development of Purkinje cells, we used inducible *Pcp2^{creER};Ai14* and *Pcp2^{creER};Kir2.1* mice to sparsely label Purkinje cells from P1 to P21 (Fig. 2a). We analyzed the dendritic arbors of Purkinje cells at P21 across different cerebellar regions. Sholl analyses revealed

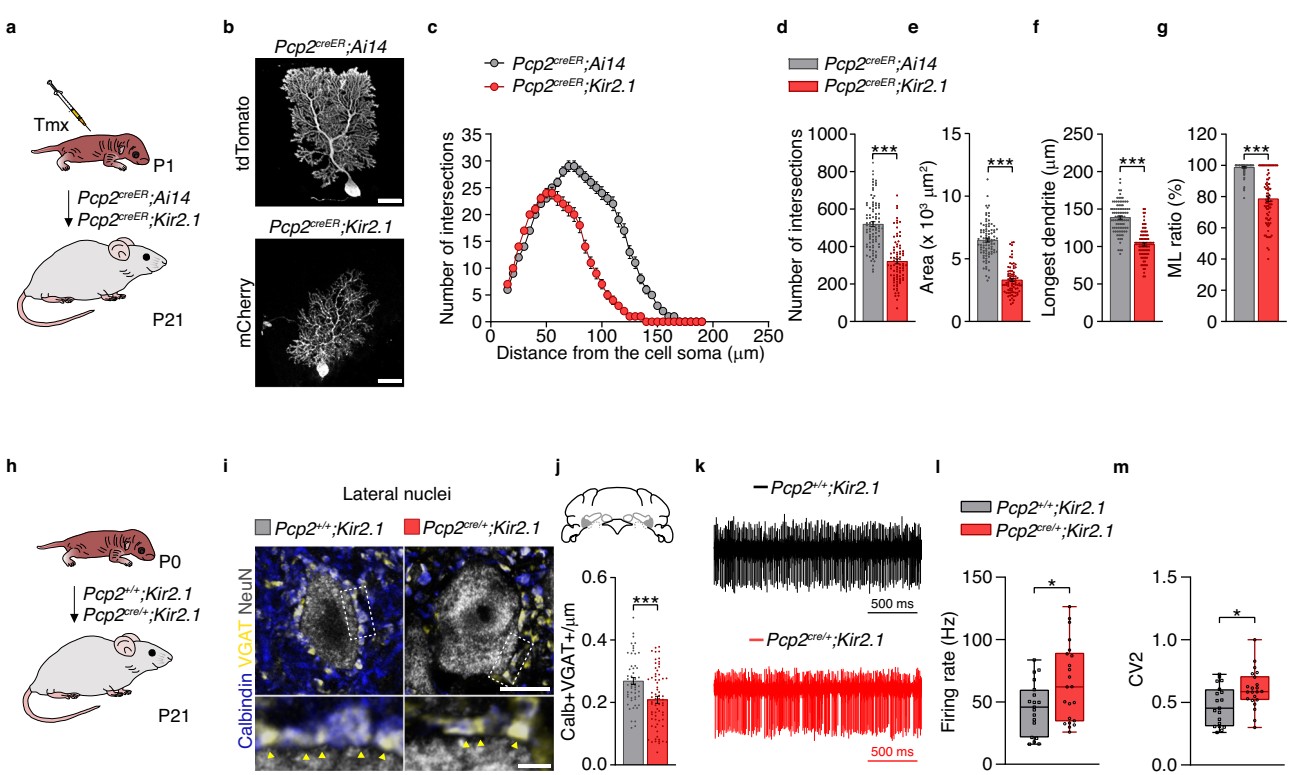

**Fig. 2 | Decreased intrinsic activity impairs Purkinje cell development.**
**a** Experimental design. *Pcp2^{creER};Ai14* and *Pcp2^{creER};Kir2.1* pups were injected with tamoxifen (Tmx) at postnatal day 1 (P1) to selectively express tdTomato or Kir2.1-mCherry in Purkinje cells, respectively. Brains were collected at P21 for morphological analyses. **b** Representative images of Purkinje cell morphology in *Pcp2^{creER};Ai14* and *Pcp2^{creER};Kir2.1* mice at P21. **c** Quantification of dendrite complexity using Sholl analysis, including **d** number of intersections, **e** cell area, **f** longest dendrite, and **g** ML ratio in *Pcp2^{creER};Ai14* (n = 92 cells/3 mice) and *Pcp2^{creER};Kir2.1* (n = 84 cells/3 mice) Purkinje cells. Unpaired Student's t-test and Mann–Whitney U-test: ***P < 0.001. **h** Experimental design. Brains from conditional *Pcp2^{cre/+};Kir2.1* and control littermate *Pcp2^{+/+};Kir2.1* mice were collected at P21. **i** Representative images (top) and high-magnification insets (bottom) showing calbindin-positive (+) (blue) Purkinje cell axon terminals and VGAT+ (yellow) presynaptic boutons surrounding

a NeuN+ (gray) neuron in the lateral cerebellar nuclei at P21. **j** Density of calbindin+ VGAT+ axon terminals contacting NeuN+ neurons at P21 in *Pcp2^{+/+};Kir2.1* (n = 56 cells/3 mice) and *Pcp2^{cre/+};Kir2.1* (n = 60 cells/3 mice). Unpaired Student's t-test: ***P < 0.001. **k** Example traces of in vivo extracellular recordings of cerebellar nuclei neurons from awake P28 ± 3 days old *Pcp2^{+/+};Kir2.1* (black) and *Pcp2^{cre/+};Kir2.1* (red) mice. **l** Firing rate and **m** coefficient of variation 2 (CV2) from *Pcp2^{+/+};Kir2.1* (n = 19 cells/4 mice) and *Pcp2^{cre/+};Kir2.1* mice (n = 22 cells/5 mice). Unpaired Student's t-test: *P < 0.05. Scale bars, 25 μm (b); 10 μm (top panel **i**); 2 μm (bottom panel **i**). Data are shown as the mean ± s.e.m. (**c–g**, **j**). Data are presented in box plots indicating the median (middle line), 25th and 75th percentiles (box), and 5th and 95th percentiles (whiskers) (**l**, **m**). Statistical details are provided in Supplementary Table 1. Source data are provided as a Source data file.

that Purkinje cells in *Pcp2^creER^;Kir2.1* mice exhibited reduced dendritic complexity compared with *Pcp2^creER^;Ai14* control cells (Fig. 2b–d). Moreover, other morphological parameters, including area, the length of the longest dendrite, and extension in the molecular layer (ML), were significantly reduced in Kir2.1-expressing Purkinje cells relative to controls (Fig. 2b, e–g). To corroborate these findings, we performed gain-of-function experiments by electroporating plasmids encoding GFP (control) and Kir2.1-T2A-tdTomato (Kir2.1) into Purkinje cell progenitors at embryonic day (E) 12.5 (Supplementary Fig. 3a). At P21, Kir2.1 Purkinje cells exhibited markedly reduced dendritic complexity compared with controls (Supplementary Fig. 3b–d), along with decreased area, longest dendritic length and extension in the ML (Supplementary Fig. 3b, e–g).

To assess whether deficits in intrinsic activity affected the presynaptic targeting of Purkinje cells as a population, we used *Pcp2^cre/+^;Kir2.1* mice and control littermates (Fig. 2h). We quantified the number of calbindin-expressing (Calb+) and vesicular GABA transporter (VGAT+) boutons contacting NeuN+ cerebellar nuclei neurons. Loss of intrinsic activity resulted in a significant reduction in the number of Calb+ VGAT+ contacts that Purkinje cells made onto lateral cerebellar nuclei NeuN+ neurons compared with controls (Fig. 2i, j). To determine how these structural changes translated into cerebellar output, we performed extracellular recordings from cerebellar nuclei neurons in vivo in *Pcp2^cre/+^;Kir2.1* mice and controls. Cerebellar nuclei were localized post hoc using Evans blue. Neurons in the cerebellar nuclei of *Pcp2^cre/+^;Kir2.1* mice displayed significantly higher firing rates (Fig. 2k, l) and increased irregularity in firing compared with controls (Fig. 2m). Overall, these results indicate that loss of Purkinje cell intrinsic activity during the early postnatal period disrupts dendritic growth and impairs presynaptic targeting.

## Disrupted intrinsic activity in Purkinje cells impairs motor performance and learning

Mutations that affect Purkinje cell development are known to cause motor impairments[26–29]. To investigate how disrupting intrinsic activity from birth to adulthood affects motor behavior, we tested adult (P60-P90) *Pcp2^cre/+^;Kir2.1* mice and control littermates, *Pcp2^+/+^;Kir2.1*. In the balance beam test, *Pcp2^cre/+^;Kir2.1* mice required more time to cross a 12 mm flat beam and made more missteps than controls (Fig. 3a, b and Supplementary Movies 1 and 2). On the accelerating rotarod, *Pcp2^cre/+^;Kir2.1* mice had a shorter latency to fall on the rotating rod at both 40 and 80 rpm compared to controls (Fig. 3c and Supplementary Movies 3). Using the LocoMouse test[30,31], we found that *Pcp2^cre/+^;Kir2.1* mice exhibited abnormal body axis swings and increased body-axis angles (Fig. 3d, e). While stride and stance distances were similar between groups, mutants took significantly longer to complete these movements (Supplementary Fig. 4a and Supplementary Movies 4 and 5). Interlimb coordination was also disrupted in the mutants, with diagonal limb pairs failing to move synchronously across different walking speeds (Fig. 4c). However, in the open field test, while the distance traveled was similar between groups, mutants moved faster than controls (Supplementary Fig. 4b). Overall, these data revealed that disruption of intrinsic Purkinje cell activity from early development to adulthood results in an ataxic phenotype characterized by impaired gait coordination and balance.

Given the importance of Purkinje cell activity for motor learning, we examined the impact of reduced intrinsic activity on cerebellum-dependent learning behaviors, such as vestibular-ocular reflex (VOR) adaptation[32,33] and eyeblink conditioning (EBC)[34,35]. We observed no differences in VOR and visual VOR gain (amplitude) and phase (timing) between genotypes (Supplementary Fig. 4e, f). However, *Pcp2^cre/+^;Kir2.1* mutants exhibited lower gain and higher phase in the optokinetic reflex (OKR) compared to controls (Supplementary Fig. 4d). In a phase-reversal VOR adaptation protocol, control animals successfully reversed their VOR direction by increasing phase over a 5-day

experiment. In contrast, *Pcp2^cre/+^;Kir2.1* mutants failed to adapt their phase or gain, indicating impaired motor learning (Fig. 3f, h and Supplementary Fig. 4g). Similarly, in the EBC paradigm, where mice learn to associate a visual conditioned stimulus (CS−LED light) with an eyeblink-inducing unconditioned stimulus (US−air puff delivered to the mouse cornea), mutant mice exhibited significantly reduced conditioned response (CR−preventive eyelid closure) percentage and amplitude compared to controls (Fig. 3g, i and Supplementary Fig. 4h, i). Importantly, both genotypes showed similar unconditioned response peak times, confirming that their ability to close the eyelid was intact (Supplementary Fig. 4j).

To further investigate the neural correlates of these behavioral deficits, we performed in vivo recordings of Purkinje cell and cerebellar nuclei neuron activity in adult mice (Supplementary Fig. 5a). Consistent with findings in juvenile animals, *Pcp2^cre/+^;Kir2.1* Purkinje cells exhibited significantly lower simple spike firing rates and larger firing irregularity than controls (Supplementary Fig. 5b–d), whereas complex spike rate and regularity were unaffected (Supplementary Fig. 5b, e, f). Unlike the juvenile mice, cerebellar nuclei neurons in adult mutant mice showed reduced firing rates with unchanged firing regularity (Supplementary Fig. 5g–i). A pathological feature shared by different subtypes of ataxia is the alteration of excitatory synaptic inputs onto the Purkinje cells, particularly, the loss of climbing fibers (CFs)[36–39] and impairment of parallel fibers (PFs)[40,41]. Using VGLUT2 as a marker for CF terminals, we found a significant reduction in the percentage of CF extension within the ML and VGLUT2 puncta density in mutants compared to controls (Supplementary Fig. 6a–d). Furthermore, the number of VGLUT2 puncta apposed to the Purkinje cell soma was increased in mutants, suggesting impaired CF translocation (Supplementary Fig. 6b, e). Examination of PF terminals using VGLUT1 revealed a marked decrease in puncta density and enlarged bouton size in mutants relative to controls (Supplementary Fig. 6f–h). Collectively, these results demonstrate that suppressing Purkinje cell intrinsic activity from early development to adulthood results in impaired motor coordination and cerebellum-dependent learning, accompanied by disrupted cerebellar firing patterns and synaptic organization.

## The requirement of intrinsic activity for Purkinje cell morphological development decreases with age

After establishing the importance of Purkinje cell intrinsic activity for dendritic growth and presynaptic targeting, we asked whether this requirement persisted across different developmental stages. To address this, we administered low doses of tamoxifen at P1, P7, or P14 in *Pcp2^creER^;Ai14* and *Pcp2^creER^;Kir2.1* mice to achieve sparse labeling in order to analyse Purkinje cell dendritic arbor morphology one week later, at P7, P14, and P21, respectively (Fig. 4a, e, i and Supplementary Fig. 7). Following Kir2.1 overexpression from P1 to P7, mutant Purkinje cells displayed a greater number of primary neurites emerging from the soma compared with controls (Fig. 4b–d). While control P7 Purkinje cells exhibited a characteristic single flattened dendrite with a vertical orientation, Kir2.1-overexpressing Purkinje cells retained an immature, multipolar morphology with neurites extending in multiple directions, resembling an earlier developmental stage[42]. Despite these differences in dendritic organization, the area and the length of the longest neurite did not differ between genotypes (Fig. 4d). Overexpression of Kir2.1 during the second postnatal week (P7−P14) caused a marked reduction in dendritic complexity, area, dendrite length, and extension in the ML in mutant Purkinje cells compared with controls at P14 (Fig. 4f–h). When intrinsic activity was suppressed during the third postnatal week (P14−P21), mutant Purkinje cells at P21 still exhibited significant reductions in dendritic complexity, area, and dendrite length relative to controls (Fig. 4j–l). However, the magnitude of these deficits−10% in dendritic complexity, 21% in area, and 13% in dendrite length−was less pronounced than that observed with activity

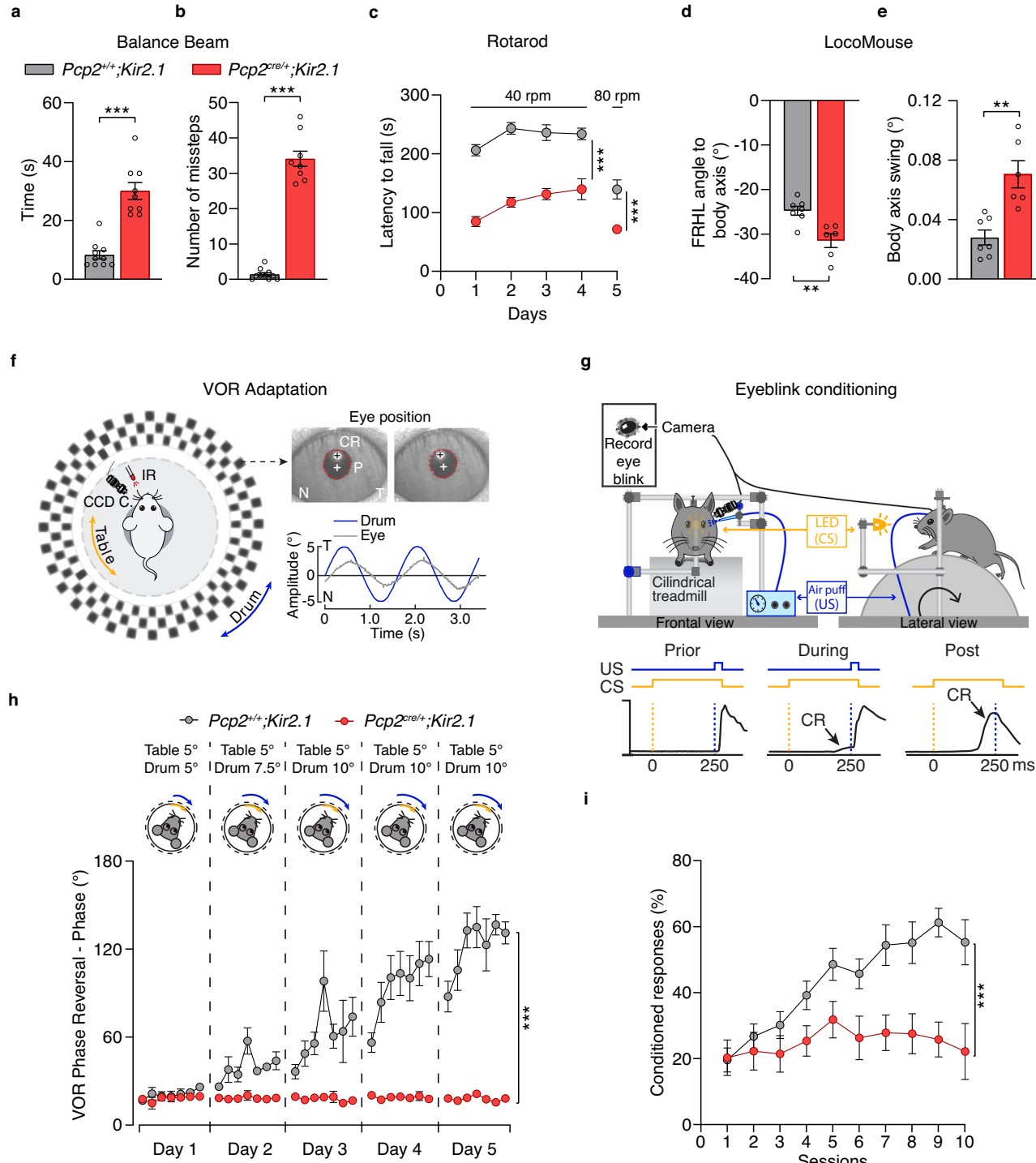

**Fig. 3 | Reduced intrinsic activity of Purkinje cells impairs balance, coordination, and cerebellar-motor learning.** Balance beam test. **a** Time to cross the beam and **b** number of missteps made by *Pcp2⁺/⁺;Kir2.1* (*n* = 10 mice) and *Pcp2ᶜʳᵉ/⁺;Kir2.1* (*n* = 9 mice). Mann–Whitney *U*-test: ***P < 0.001. **c** Accelerated rotarod performance. Latency to fall across five consecutive days in *Pcp2⁺/⁺;Kir2.1* (*n* = 11 mice) and *Pcp2ᶜʳᵉ/⁺;Kir2.1* (*n* = 10 mice). Two-way repeated measures ANOVA followed by Bonferroni's multiple comparisons: ***P < 0.001. LocoMouse gait analysis. **d** Angle of front-right hind-left (FRHL) paw placement relative to the body axis and **e** swing angle of the body axis in *Pcp2⁺/⁺;Kir2.1* (*n* = 7 mice) and *Pcp2ᶜʳᵉ/⁺;Kir2.1* (*n* = 6 mice). Unpaired Student's *t*-test: **P < 0.01. **f** Compensatory eye movement recordings. Illustration of the recording setup showing vestibular (turntable, yellow arrow) and visual (drum, blue arrow) stimulation. An infrared (IR) CCD camera tracked the left

eye (N, nasal; T, temporal). Red circles, pupil fit; black cross, corneal reflection (CR); white cross, pupil (P) center. Example trace shows eye position (gray) and drum position (blue). **g** Eyeblink conditioning. Schematic of recording setup. CS conditioned stimulus (LED light, yellow), US unconditioned stimulus (corneal air puff, blue). **h** Quantification of vestibular-ocular reflex (VOR) phase reversal training over five consecutive days in *Pcp2⁺/⁺;Kir2.1* (*n* = 7 mice) and *Pcp2ᶜʳᵉ/⁺;Kir2.1* (*n* = 7 mice). Mixed-effect analysis with repeated measures: ***P < 0.001. **i** Percentage of conditioned responses during training in *Pcp2⁺/⁺;Kir2.1* (*n* = 16 mice) and *Pcp2ᶜʳᵉ/⁺;Kir2.1* (*n* = 10 mice). Two-way repeated measures ANOVA: ***P < 0.001. Data are shown as the mean ± s.e.m. Statistical details are provided in Supplementary Table 1. Source data are provided as a Source data file.

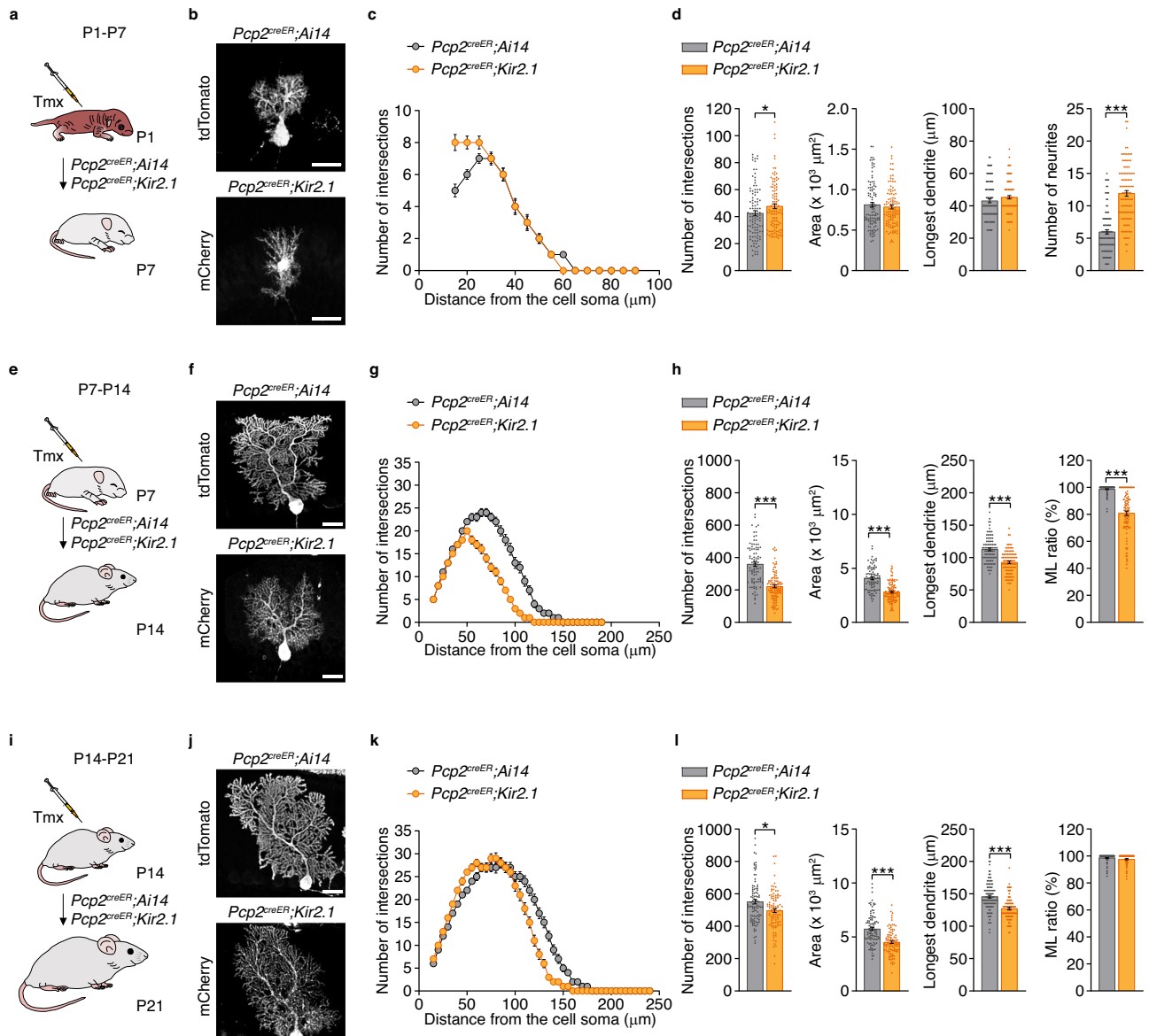

**Fig. 4 | Decreased intrinsic activity delays Purkinje cell development in the first postnatal weeks. a** Experimental design. *Pcp2^creER;Ai14* and *Pcp2^creER;Kir2.1* pups were injected with tamoxifen (Tmx) at postnatal day 1 (P1) to selectively express tdTomato or Kir2.1-mCherry in Purkinje cells, respectively. Brains were collected at P7. **b** Representative images of Purkinje cell morphology in *Pcp2^creER;Ai14* and *Pcp2^creER;Kir2.1* mice at P7. **c** Quantification of dendrite complexity using Sholl analysis, including **d** number of intersections ($P = 0.0455$), cell area ($P = 0.8336$), longest dendrite ($P = 0.0992$), and number of neurites in *Pcp2^creER;Ai14* ($n = 103$ cells/ 3 mice) and *Pcp2^creER;Kir2.1* ($n = 104$ cells/4 mice) Purkinje cells. Mann–Whitney *U*-test: *$P < 0.05$, ***$P < 0.001$. **e** Experimental design. *Pcp2^creER;Ai14* and *Pcp2^creER;Kir2.1* pups were injected with Tmx at P7, and brains were collected at P14. **f** Representative images of Purkinje cell morphology in *Pcp2^creER;Ai14* and *Pcp2^creER;Kir2.1* mice at P14. **g** Quantification of dendrite complexity using Sholl

analysis, including **h** number of intersections, cell area, longest dendrite, and ML ratio in *Pcp2^creER;Ai14* ($n = 90$ cells/3 mice) and *Pcp2^creER;Kir2.1* ($n = 101$ cells/3 mice) Purkinje cells. Mann–Whitney *U*-test and unpaired Student's *t*-test: ***$P < 0.001$. **i** Experimental design. *Pcp2^creER;Ai14* and *Pcp2^creER;Kir2.1* pups were injected with Tmx at P14, and brains were collected at P21. **j** Representative images of Purkinje cell morphology in *Pcp2^creER;Ai14* and *Pcp2^creER;Kir2.1* mice at P21. **k** Quantification of dendrite complexity using Sholl analysis, including **l** number of intersections ($P = 0.0110$), cell area, longest dendrite, and ML ratio ($P = 0.0638$) in *Pcp2^creER;Ai14* ($n = 99$ cells/3 mice) and *Pcp2^creER;Kir2.1* ($n = 95$ cells/5 mice) Purkinje cells. Mann–Whitney *U*-test and unpaired Student's *t*-test: *$P < 0.05$, ***$P < 0.001$. Scale bars, 25 μm (**b, f, j**). Data are shown as the mean ± s.e.m. Statistical details are provided in Supplementary Table 1. Source data are provided as a Source data file.

disruption throughout the entire developmental period (P1–P21: 38%, 49%, and 26%, respectively; Fig. 2c–f). Notably, at this later stage, the percentage of dendritic extension in the ML was similar between mutant and control Purkinje cells (Fig. 4l).

To confirm the effectiveness of Kir2.1-mediated suppression of intrinsic activity throughout these stages, we performed ex vivo recordings in control and Kir2.1-overexpressing Purkinje cells. In control recordings, 77%, 92%, and 81% of Purkinje cells exhibited spontaneous firing at P7, P14, and P21, respectively. In contrast, only 6%, 0%,

and 8% of *Pcp2^creER;Kir2.1* Purkinje cells were intrinsically active at the corresponding ages (Supplementary Fig. 8a, e, i). Kir2.1 overexpression also significantly altered intrinsic electrophysiological properties, including a hyperpolarized resting membrane potential (Supplementary Fig. 8b, f, j), increased hyperpolarizing currents at −140 mV (Supplementary Fig. 8c, g, k), and reduced input resistance (Supplementary Fig. 8d, h, l) across all developmental stages examined. To examine how intrinsic activity influences Purkinje cell connectivity with cerebellar nuclei neurons at different developmental stages, we

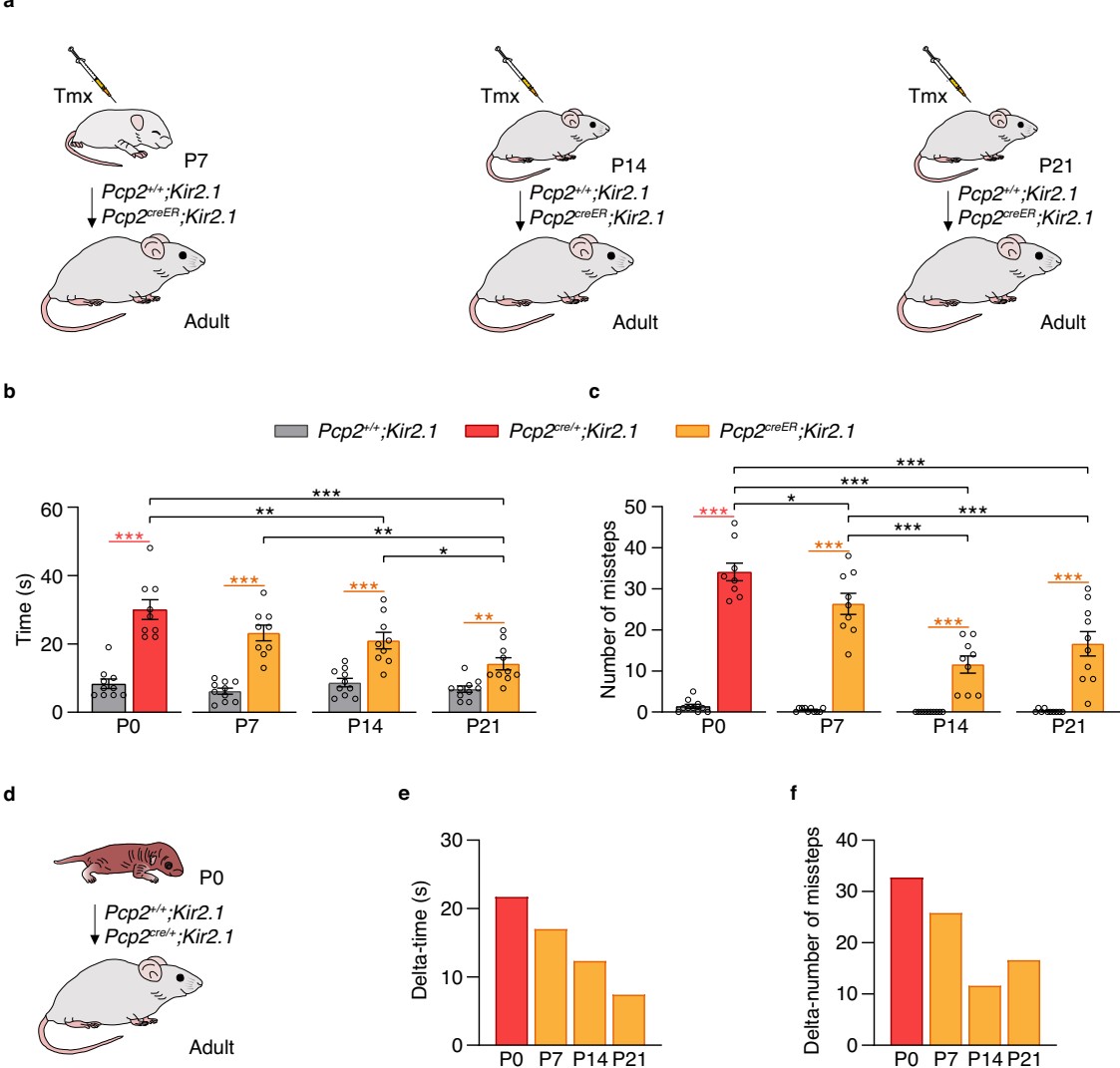

**Fig. 5 | The onset of Purkinje cell activity reduction determines the severity of motor impairment. a** Experimental design. *Pcp2^+/+;Kir2.1* and *Pcp2^creER;Kir2.1* pups were injected with tamoxifen (Tmx) at postnatal day 7 (P7), P14, or P21, and motor performance was assessed in adulthood using the balance beam test. **b** Balance beam performance showing the time to cross the beam for each induction time point in *Pcp2^+/+;Kir2.1* (*n* = 10 mice), *Pcp2^cre/+;Kir2.1* (*n* = 9 mice), and *Pcp2^creER;Kir2.1* (*n* = 9 mice). Two-sided mixed-effects analysis (*P* = 0.0011) followed by Tukey's post hoc test to correct for multiple comparisons: $^{*}P < 0.05$, $^{**}P < 0.01$, $^{***}P < 0.001$. **c** Balance beam performance showing the number of missteps for each induction time point in *Pcp2^+/+;Kir2.1* (*n* = 10 mice), *Pcp2^cre/+;Kir2.1* (*n* = 9 mice), and

*Pcp2^creER;Kir2.1* (*n* = 9 mice). Two-sided mixed-effects analysis followed by Tukey's post hoc test to correct for multiple comparisons: $^{*}P < 0.05$, $^{***}P < 0.001$. **d** Experimental design. *Pcp2^cre/+;Kir2.1* mice and littermate controls *Pcp2^+/+;Kir2.1* were assessed for motor performance in adulthood using the balance beam test. **e** Delta-time to cross the balance beam (difference in crossing time between mutant and control groups) across experimental conditions. **f** Delta-number of missteps (difference between mutant and control groups) across experimental conditions. Ages represent the onset of Kir2.1 expression in mutant animals. Data are shown as the mean ± s.e.m. Statistical details are provided in Supplementary Table 1. Source data are provided as a Source data file.

used *Pcp2^cre/+;Kir2.1* mice from P1 to P7 to achieve Kir2.1-overexpression in all Purkinje cells from birth. For the intermediary stages, P7–P14 and P14–P21, we employed the inducible *Pcp2^creER;Kir2.1* mice and administered high doses of tamoxifen to maximize Kir2.1 expression in all Purkinje cells, alongside the respective control groups (Supplementary Fig. 9a, b, e, f, i, j). Kir2.1 overexpression from P1 to P7 significantly reduced the number of Calb+ VGAT+ contacts between Purkinje cells and NeuN+ neurons in the lateral cerebellar nuclei compared with controls (Supplementary Fig. 9c, d). Suppression of intrinsic activity from P7 to P14 also led to a significant decrease in these inhibitory contacts (Supplementary Fig. 9g, h). In contrast, reducing activity from P14 to P21 had no effect on the number of Calb+ VGAT+ inhibitory contacts between Purkinje cells and NeuN+ neurons in the mutant compared with the control (Supplementary Fig. 9k, l). These findings

indicate that Purkinje cell intrinsic activity is essential for proper dendritic morphology and presynaptic targeting during development, with the most pronounced deficits occurring when activity is disrupted during the first two postnatal weeks.

## The onset of Purkinje cell intrinsic activity loss determines the severity of motor impairment

Next, we examined whether the timing of intrinsic activity disruption in Purkinje cells during development influences the severity of motor impairments. High doses of tamoxifen were administered for three consecutive days at P7, P14, or P21 in inducible *Pcp2^creER;Kir2.1* mice and respective controls to induce Kir2.1-overexpression in all Purkinje cells (Fig. 5a and Supplementary Fig. 10c-h). Motor performance in adulthood was assessed using the balance beam test. Regardless of the

timing of intrinsic activity suppression, *Pcp2^creER;Kir2.1* mice were significantly slower and made more missteps when crossing the flat 12 mm beam compared to controls, indicating impaired balance and coordination (Fig. 5b, c). However, the severity of these deficits depended on when intrinsic activity was disrupted. Mice with disrupted Purkinje cell activity from the second or third postnatal week performed significantly better than those in which Purkinje cell activity was suppressed from birth (Fig. 5b–f and Supplementary Fig. 10a, b). These findings suggest that while intrinsic Purkinje cell activity is crucial for optimizing motor movements, motor control is particularly sensitive to disruptions in Purkinje cell activity during early postnatal weeks.

Because delaying the onset of Kir2.1-overexpression by just one week was sufficient to improve balance beam performance, we hypothesized that the early deficits observed in the most severe phenotype, where Kir2.1 was overexpressed from birth (Supplementary Fig. 11a), might also impair the maturation of other cerebellar cell types whose development depends on proper Purkinje cell maturation, namely ML interneurons (MLIs)[43,44] and granule cells (GCs)[45,46]. To test this, we examined MLI density in adult *Pcp2^cre/+;Kir2.1* mice and their respective controls (Supplementary Fig. 11a, b). While parvalbumin (PV) is a known marker for both Purkinje cells and MLIs[47], it does not label all MLIs within the ML[48]. Hence, we quantified the number of PV-positive (PV+) cells co-labeled with NeuroTrace (NeuT+), a fluorescent Nissl stain, within the ML. The number of PV+ NeuT+ cells, but not NeuT+ cells, was reduced in *Pcp2^cre/+;Kir2.1* mice compared with controls (Supplementary Fig. 11b–d). The ML was also thinner than in controls (Supplementary Fig. 11b, e). We next assessed MLI density in mice in which Purkinje cell activity was disrupted at P7, thereby sparing the first week of development from changes in activity (Supplementary Fig. 11f). The density of PV+ NeuT+ and NeuT+ MLIs, as well as the ML thickness, was unchanged between groups (Supplementary Fig. 11g–j). Lastly, we analyzed the granule cell layer thickness (GCL) as a proxy for GC numbers, and measured the size of vermal cerebellar lobules in *Pcp2^cre/+;Kir2.1* mice and P7-inducible *Pcp2^creER;Kir2.1* mice and their respective controls (Supplementary Fig. 11k). There was no difference in the GCL thickness of the cerebellar cortex between *Pcp2^cre/+;Kir2.1* or P7-inducible *Pcp2^creER;Kir2.1* mice and respective controls (Supplementary Fig. 11l, n). However, lobules I-II, IV-V, and IX were smaller in *Pcp2^cre/+;Kir2.1* mice (Supplementary Fig. 11m), while lobules IV-V and VI were reduced in the P7-inducible *Pcp2^creER;Kir2.1* mice compared to controls (Supplementary Fig. 11o). Together, these results indicate that early disruption of Purkinje cell intrinsic activity not only leads to more severe motor deficits but also interferes with the proper maturation of cerebellar circuitry and structure.

### Early loss of Purkinje cell intrinsic activity activates gene networks underlying cerebellar disease

Our findings revealed that loss of Purkinje cell activity during the first week of development was sufficient to alter neuron development (Fig. 4a–d). To uncover the molecular mechanisms by which intrinsic activity influences Purkinje cell development, we isolated Purkinje cells from P7 *Pcp2^cre/+;Kir2.1* and control mice and compared their transcriptomics profiles using bulk RNA sequencing (RNAseq) (Supplementary Fig. 12a–c). Differential expression analysis identified 1602 differentially expressed genes (DEGs) (Fig. 6a). Gene ontology (GO) enrichment analysis of DEGs revealed a significant overrepresentation of biological processes related to synaptic signaling (e.g., *Grid2, Snca*), calcium ion transport (e.g., *Trpc3, Itpr1*), and regulation of nervous system development (e.g., *Hes5, Sema6d*) (Fig. 6b), reflecting the morphological and synaptic alterations observed in our model (Fig. 6b). Enriched cellular component terms included the postsynaptic density membrane, dendritic spines, presynaptic membrane and axon terminus, further supporting transcriptional dysregulation at the synaptic level (Supplementary Fig. 12d). At the molecular function

level, most DEGs were associated with transmembrane transport activity and calcium channel activity, indicating altered calcium signaling mechanisms (Supplementary Fig. 12e). These data evidence that the transcriptional changes following Purkinje cell intrinsic activity suppression, converge on pathways essential for neuronal excitability, synaptic integrity, and cerebellar circuit development. To identify functional networks affected by the loss of Purkinje cell intrinsic activity, we performed KEGG pathway enrichment analysis on the DEGs. This analysis revealed significant enrichment for pathways associated with synaptic formation and function, including axon guidance, GABAergic, dopaminergic, and glutamatergic synapses. In addition, several intracellular signaling mechanisms were enriched, such as calcium, phosphatidylinositol, and retrograde endocannabinoid signaling, as well as long-term depression, underlying the broad impact of intrinsic activity on intracellular communication and synaptic plasticity. Notably, KEGG enrichment also identified disease-related pathways, including those associated with neurodegeneration, spinocerebellar ataxia, and Huntington's disease, suggesting that early disruption of Purkinje cell activity engages molecular programs commonly implicated in cerebellar and neurodegenerative disorders (Fig. 6c). Consistent with these transcriptional alterations, loss of Purkinje cell intrinsic activity led to severe motor impairments in our mouse model. To explore disease-relevant transcriptional changes, we examined the KEGG Disease Database, which identified 216 genes linked to movement disorders with a cerebellar component, 25 of which overlapped with our DEG list (Fig. 6d and Supplementary Fig. 12f, g). These overlapping genes were key regulators of neuronal development, excitability, synaptic and mitochondrial function[49], most of which were downregulated in our dataset (Fig. 6e).

### *Prkcg* and *Car8* regulate early Purkinje cell dendritic development

To investigate whether genes implicated in cerebellar disease also contribute to Purkinje cell development, we selected two candidates from our RNA-seq analysis for functional studies: *Prkcg* (PKCγ, protein kinase C gamma)[50] and *Car8* (CAR8, carbonic anhydrase VIII)[51] (Fig. 6e). The selection criteria included: (1) fold-change in expression, (2) statistical significance, and (3) known involvement in cerebellar disease[52–54] (Supplementary Fig. 12g). We performed Purkinje cell-specific in vivo gene knockdown using effective short-hairpin RNAs (shRNAs) targeting *Prkcg* (*shPrkcg*) and *Car8* (*shCar8*), with *shLacZ* as a control (Supplementary Fig. 13a, c). Each shRNA was delivered via adeno-associated virus (AAVs) carrying an *mCherry* reporter gene (Supplementary Fig. 13e). AAVs were injected into the lateral ventricles of neonatal *Pcp2^cre/+* mice, and tissue was collected at P7 for analysis (Fig. 7a). In situ hybridization confirmed successful downregulation of *Prkcg* or *Car8* in mCherry-positive Purkinje cells (Supplementary Fig. 13b, d). Morphological analyses revealed that *Prkcg* knockdown significantly increased dendritic complexity, area, and dendritic length, while reducing the number of neurites emerging from the soma compared to controls (Fig. 7b–g). These results indicate that PKCγ acts as a negative regulator of dendritic growth. In contrast, *Car8* downregulation did not affect dendritic complexity or cell area (Fig. 7b–e). However, it reduced dendritic length and increased the number of neurites extending from the soma compared to controls (Fig. 7b, f, g), suggesting a delay in dendritic maturation similar to that observed in Kir2.1-overexpressing Purkinje cells. These findings implicate CAR8 in the transition from an immature multipolar morphology to a more mature, vertically oriented flattened dendrite.

## Discussion

Early neuronal activity is a key regulator of circuitry formation across multiple brain regions[2], yet its contribution to cerebellar maturation has remained unclear. Here, we demonstrate that the postnatal maturation, connectivity, and function of Purkinje cells are strongly

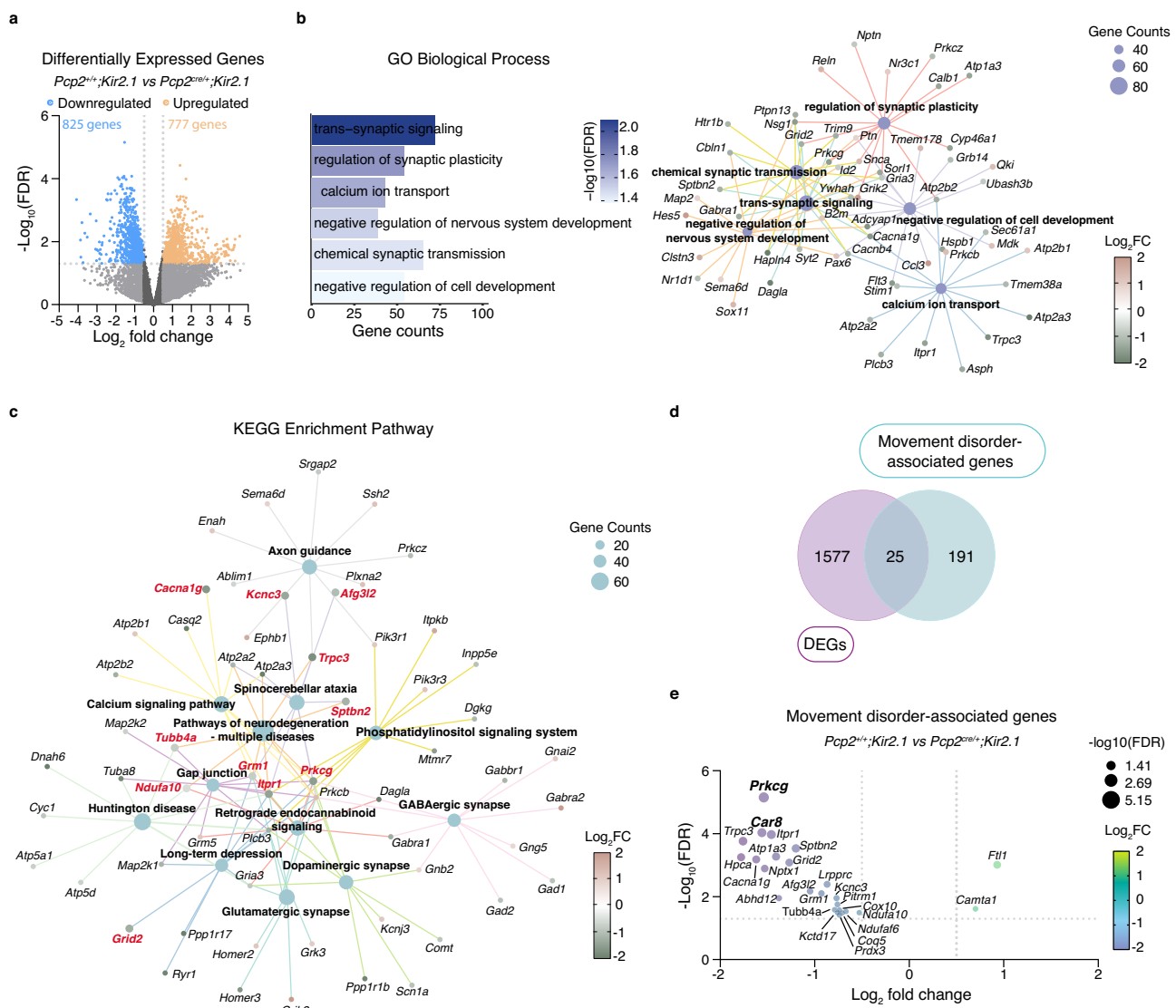

**Fig. 6 | Decreased intrinsic activity of P7 Purkinje cells alters the expression of movement disorder-associated genes. a** Vulcano plot shows DEGs in Purkinje cells isolated from *Pcp2⁺/⁺;Kir2.1* and *Pcp2ᶜʳᵉ/⁺;Kir2.1* mice at postnatal day 7 (P7). Downregulated (blue) and upregulated (orange) genes are defined by a log₂ fold change ≥ ± 0.5 and −log₁₀ (FDR) ≥ 1.3. **b** Selected significantly enriched Gene Ontology (GO) Biological Process terms (FDR ≤ 0.05) and the corresponding GO network based on DEGs in *Pcp2ᶜʳᵉ/⁺;Kir2.1* Purkinje cells at P7. **c** KEGG pathway enrichment network of DEGs in *Pcp2ᶜʳᵉ/⁺;Kir2.1* Purkinje cells at P7. Node size reflects

the number of enriched genes, and gene expression level is color-coded (pink, upregulated; green, downregulated; based on log₂ fold change, FC). **d** Venn diagram illustrates the overlap between DEGs in *Pcp2ᶜʳᵉ/⁺;Kir2.1* Purkinje cells at P7 and genes associated with cerebellar movement disorders (KEGG Disease Database). There are 25 genes common to both datasets. **e** Vulcano plot highlighting the 25 overlapping genes identified in d, plotted by log₂ fold change and −log₁₀ (FDR). Genes labeled in red in (**c**) represent a subset of these shared genes. FDR false discovery rate.

dependent on their intrinsic pacemaker activity[14]. This intrinsic firing is most crucial during the first two postnatal weeks, as disruptions during this period profoundly impair Purkinje cell development and cerebellar function. Transcriptomic profiling revealed that early Purkinje cell intrinsic activity drives the expression of activity-dependent transcriptional programs that modulate morphological development, connectivity, and motor function.

To explore the role of early activity in neuronal development and circuit formation, we used a mouse model enabling conditional overexpression of the Kir2.1 channel[23], which hyperpolarizes neurons and attenuates excitability[8,9,55]. This model exhibited a marked reduction in simple spike firing rate, and while Purkinje cells were virtually silent, they remained responsive to injected current or climbing fiber activity, retaining their ability to fire action potentials. Reducing neuronal activity delayed Purkinje cell dendritic maturation. After one week of

neuronal silencing during the first postnatal week, Purkinje cells resembled what Ramón y Cajal described as stellate cells with disoriented dendrites[56]. This stage is part of normal dendritic development and is characterized by multiple perisomatic protrusions emerging from the soma, in contrast to the typical Purkinje cell morphology, where a small, flattened apical dendrite extends from the soma. A comparable phenotype has been reported in Autism Susceptibility Candidate 2 (*Auts2*) deficient mice[57]. *Auts2* regulates multiple developmental processes, and its mutation is linked to autism spectrum disorders, intellectual disability, schizophrenia, and epilepsy[58]. Temporal control of activity allowed us to define a sensitive developmental period. Perturbing intrinsic activity only in the third postnatal week did not reproduce the severe morphological defects observed when activity was suppressed throughout the first three postnatal weeks, suggesting that Purkinje cells undergo activity-

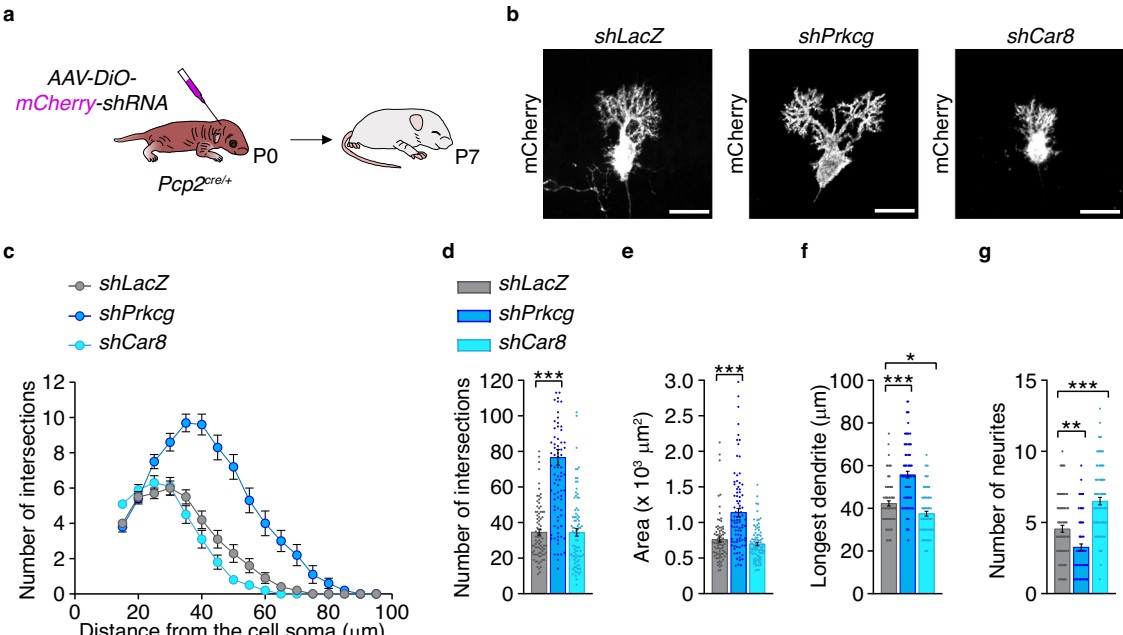

**Fig. 7 | Activity-dependent Prkcg and Car8 genes regulate Purkinje cell differentiation at P7. a** Experimental design. *Pcp2*[cre/+] pups were injected at postnatal day 0 (P0) with a cre-dependent adeno-associated vector (AAV) vector carrying short hairpin RNA (*shRNA*) against *LacZ* (control), *Prkcg*, or *Car8*, along with an *mCherry* reporter gene. Brains were collected at P7. **b** Representative images of Purkinje cell morphology in *shLacZ*-, *shPrkcg*-, and *shCar8*-injected mice at P7. Scale bars, 25 μm. **c** Quantification of dendrite complexity using Sholl analysis, including **d** number of intersections, **e** cell area, **f** longest dendrite, and **g** number of neurites in *shLacZ* (n = 84 cells/4 mice), *shPrkcg* (n = 90 cells/3 mice), and *shCar8* (n = 90 cells/3 mice) Purkinje cells. Kruskal–Wallis followed by Dunn's multiple comparisons: *P < 0.05, **P < 0.01, ***P < 0.001. DiO double-floxed inverted orientation. Data are shown as the mean ± s.e.m. Statistical details are provided in Supplementary Table 1. Source data are provided as a Source data file.

dependent differentiation and maturation, processes that are particularly relevant in the first two postnatal weeks. This aligns with and expands on previous findings that early postnatal development is critical for most cerebellar maturation processes[11]. For instance, delaying the expression of the mutant *Ataxin1* gene until after cerebellar development markedly reduces disease severity, Purkinje cell loss, and motor impairment[59,60]. Similarly, in a Purkinje cell-specific deletion for *Tsc1* autism model, rapamycin treatment during development prevented Purkinje cell death, rescued social deficits[61], and restored Purkinje cell intrinsic activity[62]. We also found that loss of intrinsic activity reduced the number of presynaptic inhibitory contacts from Purkinje cells onto cerebellar nuclei neurons, an activity- and time-dependent effect absent when activity was suppressed only in the third postnatal week. These structural changes may reflect impaired axon development, failure of synapse formation, elimination of weak or non-functional synapses, and/or defective maintenance[63]. Consistent with this, our RNA-seq data indicated that intrinsic activity regulates genes involved in axon guidance and synapse formation/maintenance. Specifically, *Sema6d* promotes retinal axon midline crossing[64] and is required for synapse formation and GABA transmission in the amygdala[65]. *Srgap2*, also differentially expressed in silent Purkinje cells, induces neurite outgrowth and branching[66] and promotes synapse formation in neocortical neurons[67].

Our results highlight the first two postnatal weeks as a sensitive period during which perturbations of Purkinje cell intrinsic activity have profound consequences. During this time window, Purkinje cells are transiently interconnected via axon collaterals, generating early spontaneous network activity driven by intrinsic activity[68], while other circuit components are still immature. At this developmental stage, the GABA equilibrium potential is depolarizing and switches to inhibitory, a switch that occurs around P8/9 in the rat[69]. In this depolarizing phase, GABA$_A$ receptors mediate calcium transients in immature Purkinje

cells[69], and this time window overlaps with the presence of transient traveling waves propagated through Purkinje cell axon collaterals, which depend on GABA$_A$-receptor-mediated transmission. By the third postnatal week, axon collaterals are largely pruned, and the transient waves disappear[68]. In the first postnatal week, our RNA-seq data show that Purkinje cell intrinsic activity activates a molecular program regulating trans-synaptic signaling, neurodevelopment, and calcium transport. Several calcium signaling genes are activity-dependent (*Cacna1g*, *Grm1*, *Hpca*, *Trpc3*, *Camta1*, *Car8*, *Itpr1*, *Prkcg*). Before synaptic networks are fully mature, calcium signaling serves as the primary source of activity regulating multiple aspects of neuronal development, including dendritic growth and patterning, axonal growth and pathfinding, and neurotransmitter specification[70]. Importantly, disrupted calcium homeostasis is a hallmark of many cerebellar movement disorders[12,71,72], and such disruptions are known to impair Purkinje cell intrinsic firing[73]. Our dataset also revealed activity-dependent regulation of genes involved in mitochondrial function (*Ndufa10*, *Prdx3*, *Lrpprc*, *Afg3l2*, *CoqS*, *Pitrm1*, *Cox10*, *Nduf6*). Mitochondrial metabolism is essential for calcium buffering[74], neurodevelopment[75], neuronal excitability[76], and synaptic transmission[77], and its impairment has been implicated in cerebellar growth arrest[78] and cerebellar ataxia[79]. During this early stage of development, MLIs are still dividing the white matter and begin migrating into the ML[80], processes that are partially regulated by the Purkinje cells[44,81]. The loss of Purkinje cell intrinsic activity during the first postnatal week did not change the number of MLIs in the adult ML, but it reduced the levels of PV+ MLIs. Parvalbumin, a calcium-binding protein, is a marker of MLI maturation[82] and contributes to MLI presynaptic calcium signaling[83]. We therefore propose that Purkinje cell intrinsic activity is required for MLI maturation, either through activity-dependent gene expression, Purkinje cell-mediated paracrine signaling, or via remodulation of the excitatory drive, as the number of

VGLUT1-positive parallel fiber inputs is also reduced in our mouse model. During the second postnatal week, CFs transition from multi-innervation of the Purkinje cell soma to elimination, translocation, and strengthening of the remaining synapses on proximal dendrites[84]. In addition, PFs begin to innervate distal dendrites[85] while most of the MLIs start to integrate into the nascent circuits[86]. By the third postnatal week, most Purkinje-Purkinje cell synapses have disappeared, and excitatory synaptic inputs reach mature numbers and strength. In vitro recordings show that by the end of the second postnatal week, intrinsic firing frequencies approximate those of adult cells and remain stable thereafter[21]. Together, we posit that the immature cerebellum undergoes extensive morphological and physiological remodeling, which relies heavily on early Purkinje cell intrinsic activity mediated via GABA-evoked calcium signaling, before the maturation of synaptic inputs. Disrupting intrinsic activity during this early developmental window is sufficient to impair the emergence of a mature, functional cerebellum.

A common feature across multiple mouse models of disease, including cerebellar ataxia[87], Huntington's disease[88], autism spectrum disorder[89], and neonatal brain injury[90], is the disruption of Purkinje cell intrinsic activity. Cook et al. proposed that deficits in Purkinje cell intrinsic firing, even during asymptomatic stages, can drive the onset of cerebellar disease[12]. According to this, restoring the levels of Purkinje cell intrinsic activity with 3,4-diaminopyridine in a mouse model of spinocerebellar ataxia 1 (SCA1), reduced the locomotion phenotype[87]. In our mouse model, loss of intrinsic activity from birth was sufficient to impair balance, coordination, and cerebellar-dependent motor learning. These findings align with deficits reported after repeated periods of hypoxia during the first two postnatal weeks, which resulted in persistent cerebellar deficits[90,91]. Neural activity preceding synapse formation can be disrupted by prenatal drug exposure, maternal ethanol consumption, perinatal hypoxia or ischemia, bacterial or viral infections, all of which have been linked to neurological disorders such as epilepsy and sensory and cognitive impairments[92]. However, it remains to be determined whether early perturbation of Purkinje cell intrinsic activity is sufficient to induce specific cognitive and emotional deficits. A consequence of disrupting Purkinje cell intrinsic activity was a reduction in the number of inhibitory inputs onto cerebellar nuclei neurons, accompanied by altered firing activity within the cerebellar nuclei. In juvenile mice (~P28), the firing frequency of cerebellar nuclei neurons was increased following the loss of Purkinje cell intrinsic activity. In adult mutants, this firing frequency was instead reduced. An increased firing rate in cerebellar nuclei neurons has been reported in models of cerebellar ataxia[93,94], whereas other mouse models have shown reduced firing frequency, such as in SCA3 mice[95] and in models in which Purkinje cell neuro-transmission is blocked in cerebellar neurons with a decrease in regularity[18]. It remains unclear whether the reduction in firing frequency from juvenile to adult reflects the natural progression of the disease, circuit-level changes that emerge over time, or compensatory mechanisms such as changes in synchrony, shifts in synaptic strength, or homeostatic plasticity within cerebellar nuclei circuits[96].

Loss of Purkinje cell activity after the cerebellar developmental period still resulted in motor impairments; however, the severity of these deficits was markedly reduced. Delaying the onset of Purkinje cell dysfunction by just one week after birth was sufficient to lessen impairments in the adult balance beam test, and this improvement became even more pronounced when intrinsic Purkinje cell activity remained unperturbed for two or three weeks after birth. These findings align with previous work showing that postponing mutant ATXN1 expression until after cerebellar development is complete substantially reduces disease severity in adulthood[59,60]. Altogether, these data indicate that motor skills depend on the establishment of precise nascent circuits during cerebellar development, and that the earlier

circuit assembly is compromised, the more severe the resulting phenotype.

Cerebellar-dependent motor learning is supported by several forms of synaptic and non-synaptic plasticity related to the excitatory input from CFs, PFs, and that of the inhibitory ML interneurons[97,98]. In this study, loss of Purkinje cell intrinsic activity from birth until adulthood impaired the acquisition of cerebellar-dependent motor learning. Examination of excitatory afferents revealed that VGLUT2-positive climbing fiber terminals remained clustered around the Purkinje cell soma, indicating a failure of normal climbing fiber elimination. This phenotype is consistent with previous work showing that postnatal expression of a chloride channel in Purkinje cells, which selectively reduces their excitability during synapse elimination, disrupts climbing fiber pruning. Likewise, experimentally decreasing Purkinje cell excitability has been shown to prevent the elimination of supernumerary climbing fiber inputs[99]. We also observed deficits in parallel fiber presynaptic terminals, including both reduced terminal number and enlarged bouton size. A late phase of climbing fiber synapse elimination relies first on interactions between PFs and the glutamate receptor delta 2 subunit (GluRδ2)[41] and subsequently on activation of the metabotropic glutamate receptor 1 (mGluR1) signaling pathway at distal Purkinje cell dendrites[100]. Consistent with this, both Grid2 (encoding GluRδ2) and Grm1 (encoding mGluR1) were downregulated in mutant Purkinje cells at P7, and mutations in these genes have been identified in patients with cerebellar ataxia[101,102]. Additionally, deficits were detected in the climbing fiber input organization, a feature commonly observed in spinocerebellar ataxias[103]. Overall, the loss of intrinsic Purkinje cell activity profoundly disrupted excitatory synaptic maturation and, ultimately, cerebellar-dependent motor learning.

Many of the genes identified during the first postnatal week as differentially expressed in the absence of Purkinje cell intrinsic activity have previously been implicated in cerebellar development and cerebellar disorders. Among these, Prkcg and Car8 are of particular interest, as both are associated with cerebellar dysfunction and exhibited altered expression in our model. Prkcg encodes for protein kinase C gamma (PKCγ), a serine/threonine kinase highly expressed in Purkinje cells[104] and involved in several key neuronal processes such as synaptic maturation[105], and regulation of long-term depression[106]. Mutations in Prkcg cause Spinocerebellar ataxia 14 (SCA14), characterized by gait imbalance, dysarthria, and abnormal eye movements[52,107]. Notably, SCA14 pathology can arise through distinct mechanisms depending on the specific mutation and its impact on PKCγ function, with both gain-of-function[108,109] and loss-of-function[110] alterations reported[111]. In vitro studies have shown that PKCγ activation inhibits Purkinje cell dendritic growth, whereas its inhibition promotes dendritic expansion[112]. These findings are in accordance with our in vivo functional analysis, in which downregulation of Prkcg increased dendritic complexity, a result that contrasts with the stunted dendritic growth in our mouse model. These data suggest that Prkcg acts as a negative regulator of excessive dendritic growth during development, a role that may be dysregulated in SCA14. The carbonic anhydrase-related protein 8 (CAR8) is an acatalytic member of the carbonate dehydratase family, and it is highly expressed in Purkinje cells[51]. Car8 mutations cause cerebellar ataxia with intellectual disability and disequilibrium syndrome[53,113]. Loss-of-function mouse models for CAR8 exhibit ataxia and dystonia, while they do not display gross morphological defects in adulthood; transient anatomical alterations are known to precede motor dysfunction[114,115]. In our study, Car8 downregulation closely mirrored the altered dendritic development observed in the mutant Purkinje cells. These findings highlight the role of CAR8 in early Purkinje cell maturation, specifically in promoting the retraction of excess neurites in immature Purkinje cells and establishing dendritic branches. Together, these gene-specific insights align with our broader conclusion that Purkinje cell intrinsic activity plays a central role in orchestrating

early cerebellar development through activity-dependent transcriptional programs that govern Purkinje cell maturation, connectivity, and ultimately motor function. The convergence of developmental and adult cerebellar disease biological pathways, including calcium and phosphatidylinositol signaling, as well as synaptic maintenance mechanisms, supports the hypothesis that late-onset disorders may have roots in earlier neurodevelopmental events[116,117]. Given the extended timeline of cerebellar maturation[11], exploring the developmental origins of adult cerebellar disease represents an intriguing, although challenging, direction for future research.

## Methods

### Mouse strains
The following transgenic mouse lines were used in this study and maintained on a C57BL/6 background (Charles River Laboratories): *Pcp2^creER* (*Tg(Pcp2-creER^{T2})17.8.ICS*)[22], *Ai14* (*B6;129S6-Gt(ROSA) 26Sor^{tm14(CAG-tdTomato)Hze}/J*)[24], *Kir2.1* (*Gt(ROSA)26Sor^{tm2(CAG-KCNJ2/mCherry)Fmr}*)[23] and *Pcp2^cre/+*(*B6.Cg-Tg(Pcp2-cre)3SSSJdhu/J*)[118]. *Ai14* (Jackson Laboratories #007908) and *Kir2.1* (kindly provided by Guillermina López-Bendito) mouse lines were maintained as homozygous and crossed with cre-driver lines, resulting in progeny that were always heterozygous for *Ai14* and *Kir2.1*. *Ai14* and *Kir2.1* mice were crossed with inducible *Pcp2^creER* mice to drive tdTomato or Kir2.1-mCherry expression in Purkinje cells, respectively, following tamoxifen administration. Tamoxifen (Sigma-Aldrich, Cat. #T5648, 20 mg/ml in corn oil) was administered subcutaneously or intraperitoneally to induce Cre recombination. The *Ai14* and *Kir2.1* lines were also crossed with *Pcp2^cre/+* mice to conditionally express tdTomato or Kir2.1 in all Purkinje cells, respectively. Both male and female mice were used in all experiments. For *in utero* electroporation, time-mated FVB/NHsd (Envigo) female mice were used and housed individually. The day of the vaginal plug detection was designated as embryonic day (E) 0.5. All animals were maintained under standard temperature-controlled laboratory conditions on a 12 h:12 h light/dark cycle, with water and food available *ad libitum*. All procedures were approved by the Dutch Ethical Committee for Animal Experiments and conducted in accordance with the Institutional Animal Care and Use Committee (IACUC) of Erasmus Medical Center, and with the European and Dutch National Legislation.

### Tamoxifen protocols
Tamoxifen dosing and timing were adjusted to achieve either sparse or complete Purkinje cell labeling at different developmental stages, driving the expression of tdTomato or Kir2.1-mCherry, in *Pcp2^creER;Ai14* and *Pcp2^creER;Kir2.1*, respectively. Protocols were designed with consideration of the developmental onset and pattern of *Pcp2* expression. Sparse labeling, which allows visualization of individual cell morphology, was achieved by administering tamoxifen at 100 mg/kg body weight at postnatal day (P) 1.5 mg/kg at P7, and 1 mg/kg at P14. For recombination in all Purkinje cells, in short-term studies, 100 mg/kg tamoxifen was administered at P7 and 50 mg/kg at P14 to label Purkinje cells over a defined one-week window. For long-term recombination extending into adulthood, *Pcp2^creER;Kir2.1* mice and their control littermates received tamoxifen at 100 mg/kg for three consecutive days at P7, P14, or P21. To achieve recombination in all Purkinje cells from birth while avoiding multiple injections in neonatal pups, constitutive *Pcp2^cre/+;Kir2.1* mice and their control littermates were used, ensuring expression of tdTomato or Kir2.1-mCherry in all Purkinje cells from the earliest developmental stages. An overview of all experimental designs is provided in Supplementary Fig. 14.

### Ex vivo Purkinje cell recordings and analysis
Ex vivo slice recordings of Purkinje cells were performed on cerebellar slices obtained from a total of 52 mice at three developmental ages: P7, P14, and P21 (±1 day). Cerebellar tissue was collected from *Pcp2^creER;Ai14* or *Pcp2^creER;Kir2.1* mice that had been injected with

tamoxifen at P1, P7, or P14 to induce sparse expression of tdTomato or Kir2.1-mCherry, respectively, in Purkinje cells. Brains were rapidly removed and placed in an ice-cold slicing solution continuously equilibrated with 95% $O_2$ and 5% $CO_2$, containing (in mM): 240 Sucrose, 2.5 KCl, 1.25 $NaH_2PO_4$, 2 $MgSO_4$, 1 $CaCl_2$, 26 $NaHCO_3$, and 10 D-glucose. Sagittal vermal cerebellar slices (250 μm thick) were obtained in ice-cold slicing solution using a vibratome (VT1000S, Leica Biosystems, Wetzlar, Germany) equipped with a ceramic blade (Campden Instruments Ltd, Manchester, United Kingdom). Slices were then incubated in oxygenated artificial cerebrospinal fluid (aCSF) at 34 °C for 1 h before being transferred to a recording chamber. The chamber was maintained at 34 °C using a feedback temperature controller (Scientifica, Uckfield, United Kingdom) and continuously perfused with oxygenated aCSF containing (in mM): 124 NaCl, 5 KCl, 1.25 $Na_2HPO_4$, 2 $MgSO_4$, 2 $CaCl_2$, 26 $NaHCO_3$, and 20 D-glucose[21].

For all the recordings, the aCSF was supplemented with synaptic receptor blockers to isolate the intrinsic activity of Purkinje cells. The following antagonists were included: the NMDA receptor antagonist D-AP5 (50 μM), the selective and competitive AMPA receptor antagonist NBQX (10 μM), and the non-competitive $GABA_A$ receptor antagonist and glycine receptor inhibitor picrotoxin (100 μM; all from Hello Bio Ltd, Bristol, United Kingdom). For some recordings, the aCSF was additionally supplemented with 300 μM barium (Sigma-Aldrich, Merck KGaA, Darmstadt, Germany). Purkinje cells were selected based on their fluorescent labeling: tdTomato expression in *Pcp2^creER;Ai14*, absence of mCherry in *Pcp2^creER;Kir2.1,Ctl*, and mCherry expression in *Pcp2^creER;Kir2.1*. Neurons were visualized using a SliceScope Pro 3000 microscope, with a CCD camera, a trinocular eyepiece (Scientifica, Uckfield, UK), and an ocular (Teledyne Qimaging, Surrey, Canada).

Whole-cell recordings were obtained using borosilicate pipettes (Harvard apparatus, Holliston, MA, USA) with a resistance of 4−6 MΩ, filled with an internal solution containing (in mM): 9 KCl, 3.48 $MgCl_2$, 4 NaCl, 120 $K^+$-Gluconate, 10 HEPES, 28.5 Sucrose, 4 $Na_2ATP$, and 0.4 $Na_3GTP$, adjusted to pH 7.25-7.35 and osmolarity 290−300 mOsmol/Kg (Sigma-Aldrich, Merck KGaA, Darmstadt, Germany). Whole-cell recordings were performed on Purkinje cells held at −65 mV. Before establishing the whole-cell recording configuration, capacitance was optimally compensated. Input resistance was determined using the seal test protocol integrated into PatchMaster software (HEKA Elektronik). Immediately after achieving the giga-ohm seal (>1 GΩ), a test pulse was applied through PatchMaster (a 10 mV hyperpolarizing voltage step for 10−20 ms) to assess passive membrane properties without activating voltage-gated currents. Membrane resistance ($R_m$) was calculated by subtracting the series resistance ($R_s$) from the total input resistance ($R_{in}$) ($R_m = R_{in} − R_s$). Recordings with a series resistance greater than 25 MΩ were excluded. After achieving whole-cell configuration, zero current was injected in current-clamp mode to assess intrinsic action potential firing and resting membrane potential, defined as the mode of the voltage trace. Intrinsic excitability was tested by holding the cells at −65 mV using a holding current, which was then labeled as '0 pA'. From this holding current, 500 ms current steps increasing by 100 pA for every step were applied, and the number of action potentials in the resulting voltage trace was counted. The current−voltage (I−V) curves were generated in voltage-clamp mode by applying 500 ms voltage steps from −140 mV to 0 mV in 10 mV increments and measuring the mode of the resulting current. Recordings were acquired with Patchmaster (HEKA Electronics, Lambrecht, Germany) using an ECP-10 amplifier (HEKA Electronics, Lambrecht, Germany) and digitized at 20 kHz. Data files were imported into and analyzed with a custom-built MATLAB code (Mathworks, Natick, MA, USA).

### In vivo extracellular recordings and analysis
In vivo extracellular recordings of Purkinje cells and cerebellar nuclei neurons were performed in juvenile (P28 ± 3 days; *n* = 10) and adult

(P60–P90) ($n = 9$) $Pcp2^{cre/+}$;$Kir2.1$ mice and their control littermates. Mice were anesthetized with isoflurane in oxygen (4% for induction and 1.5–2% for maintenance). The scalp was shaved and incised along the rostrocaudal midline, and the medial neck muscles overlaying the occipital bone were removed. Next, a craniotomy was made in the occipital bone using a high-speed diamond-tipped drill (Kulzer, #H71104003). A custom-made recording chamber (Charisma Flow Row composite, #66095845) was positioned around the craniotomy and temporarily sealed with bonewax (Ethicon, W30) to protect the exposed brain. Then, a custom-built metal pedestal with a square magnet was fixed to the frontal and parietal bones to allow for head fixation during recordings. Mice were allowed to recover for three days post-surgery. On the day of recording, mice were head-fixed in a mouse holder, and the bonewax was removed to expose the cerebellar surface for extracellular recordings[25].

Extracellular activity was recorded using borosilicate glass pipettes (World Precision Instruments, BF200-116-10) pulled to have a long, thin taper and a resistance of 2–4 MΩ, filled with 2 M NaCl solution. One silver chloride electrode wire was inserted into the pipette to record electrical activity, and a second silver chloride electrode wire was placed in the saline-filled recording chamber as a reference electrode. Pipettes were advanced into the cerebellar cortex or nuclei using a micromanipulator (SM-5, Luigs & Neumann, Ratingen, Germany). Signals were pre-amplified (custom-made preamplifier, 1000× DC), filtered (CyberAmp 320, Axon Instruments, Molecular Devices, Sunnyvale, CA, USA), digitized (Power1401, CED, Cambridge, UK), and stored for offline analysis. To histologically verify the recording site of cerebellar nuclei neurons, at the end of some recording sessions, 0.5% Evans Blue solution was injected at the recording location with a micropipette (Hirschmann, Z611239-250EA). Recordings from Purkinje cells and cerebellar nuclei neurons were analyzed using SpikeTrain software (Neurasmus BV, Rotterdam, The Netherlands), running in MATLAB 2014a (MathWorks, Natick, MA, USA). SpikeTrain employs wave-clustering algorithms to discriminate simple and complex spikes. Purkinje cells were identified by the presence of both simple and complex spikes and confirmed as single units by the characteristic pause in simple spike firing following each complex spike. For each neuron and spike type, the firing rate and the coefficient of variation (CV) 2 were calculated. The CV2 measures local, spike-to-spike variability in firing pattern and was calculated as $2 \times |ISI_{n+1} - ISI_n|)/(ISI_{n+2}/ISI_n)$[119], where ISI = inter-spike intervals.

## Immunohistochemistry

Mice were deeply anesthetized with sodium pentobarbital by intraperitoneal injection and transcardially perfused with sodium chloride solution, followed by 4% paraformaldehyde (PFA) in 0.1 M phosphate buffer (PB; pH 7.4). Dissected brains were post-fixed for 2 h at 4 °C in 4% PFA and cryoprotected overnight at 4 °C in 30% sucrose/0.1 M PB. The next day, brains were embedded in 14% gelatin/ 30% sucrose/0.1 M PB solution, fixed for 2 h at room temperature, and incubated overnight at 4 °C in 30% sucrose/0.1 M PB. Sagittal or coronal brain sections (50 μm) were cut using a freezing microtome.

Free-floating sections were rinsed in 0.1 M PB and subjected to antigen retrieval by incubation for 2 h in 10 mM sodium citrate solution at 80 °C. Sections were then rinsed and incubated for 2 h at room temperature in a blocking solution containing 0.5% Triton X-100 and 10 % normal horse serum in 0.1 M PB to prevent nonspecific binding. Subsequently, sections were incubated overnight at 4 °C with primary antibodies diluted in antibody solution (0.5 % Triton X-100 and 2 % normal horse serum in 0.1 M PB).

The following primary antibodies were used: rabbit anti-RFP (1:1000, Rockland, #600-401-379), chicken anti-RFP (1:1000, Rockland, #600-901-379), goat anti-GFP (1:1000, Rockland, #600-101-215), mouse anti-calbindin D-28K (1:10000, Swant, #CB300), guinea pig-VGAT (1:500, Synaptic Systems, #131004), guinea pig-VGLUT1 (1:2000, Merck, AB5905), guinea pig-VGLUT2 (1:2000, Merck, #AB2251-I), mouse-parvalbumin (1:5000, ThermoFisher Scientific, MA5-47410), and rabbit-NeuN (1:1000, Millipore, #ABN78).

Sections were rinsed in 0.1 M PB and incubated for 2 h at room temperature with secondary antibodies: Cy3-AffiniPure Donkey anti-Rabbit (1:1000, #711-165-152), Cy3-AffiniPure Donkey anti-Chicken (1:1000, #703-165-155), Alexa Fluor 488-AffiniPure Donkey anti-Goat (1:1000, #705-545-147), Alexa Fluor 488-AffiniPure Donkey anti-Mouse (1:1000, #715-545-150), Cy3-AffiniPure Donkey anti-Mouse (1:1000, #715-165-150), Alexa Fluor 488-AffiniPure Donkey anti-Guinea Pig (1:1000, #706-545-148), Alexa Fluor 647-AffiniPure Donkey Anti-Mouse (1:1000, #715-605-151), and Cy5-AffiniPure Donkey anti-Rabbit (1:1000, #711-175-152) all from Jackson ImmunoResearch.

Finally, sections were counterstained with the fluorescent nuclear dye 4′,6-diamidino-2-phenylindole (DAPI, Thermo Scientific, #D3571) and mounted with Mowiol (Polysciences, Inc., #17951). Sections used for ML interneuron quantification were counterstained with Neuro-Trace 435/455 Blue Fluorescent Nissl Stain (1:300, ThermoFisher Scientific, N21479).

## Image acquisition and analysis

Images were acquired at a resolution of 1024 × 1024 pixels and 8-bit depth using either an LSM 700 or LSM 900 confocal laser scanning microscope, and an Axio Imager.M2 epifluorescence microscope (Carl Zeiss Microscopy, LLC, USA). For each experiment, all imaging parameters, including laser power, detection filter settings, pinhole size, and photomultiplier gain, were kept constant across samples.

To image individual Purkinje cells, sagittal sections were acquired using a 20×/0.8 NA or 40×/1.3 NA oil-immersion objective with a z-step of 1 μm. Given the morphological variability of Purkinje cells across cerebellar regions[21], cells were imaged throughout the cerebellar cortex and plotted individually. Dendrite complexity was quantified from maximum intensity projections of z-stack images using the Sholl analysis macro in FIJI (ImageJ) software[120]. To quantify Purkinje cell area excluding the axon, the maximum projection of each image was thresholded in FIJI, and the area was then measured. The longest dendrite length was obtained as a result of Sholl analysis. The ML ratio was calculated as the percentage of Purkinje cell height (the distance from the top of the Purkinje cell soma to the apical edge of the neuron) divided by the ML height. Neurite number was measured as the number of processes originating from the soma of a Purkinje cell.

To image calbindin-positive (+) and VGAT+ presynaptic puncta, coronal cerebellar sections were acquired using a 63×/1.4NA oil-immersion objective with a z-interval of 1 μm. For quantification of presynaptic densities, the perimeter of NeuN+ cerebellar nuclei neuron soma was measured using FIJI. Presynaptic puncta calbindin+ and VGAT+ were automatically detected using the Find Maxima function to count the number of puncta colocalizing with the soma perimeter. The number of calbindin+ signal colocalizing with VGAT+ puncta was normalized to the corresponding soma perimeter of each analyzed cell. Given the diversity in soma size among cerebellar nuclei neurons[121,122], data were plotted individually.

Tile-scan images of Purkinje cells and VGLUT2 puncta were acquired with a 40×/1.3NA oil-immersion objective with a z-interval of 0.8 μm. Maximum intensity projections of sagittal sections were used for quantification. The ML thickness was measured as the distance from the top of a Purkinje cell soma to the apical edge of the ML. Climbing fiber (CF) height was measured as the distance from the top of a Purkinje cell soma to the most distal VGLUT2 puncta. ML thickness and CF height were determined from three measurements per image. CF extension was quantified as the percentage of the CF height relative to ML thickness. VGLUT2 density was quantified by counting VGLUT2 puncta within a specific region of interest (ROI) using the 'Analyze Particles' function in FIJI, and normalizing the count to the ROI area. To quantify VGLUT2 perisomatic puncta, the perimeter of individual

Purkinje cell soma was measured, and VGLUT2 puncta colocalizing with the soma perimeter were counted, excluding those inside the soma. Measurements were performed from three Purkinje cells per image, and the number of VGLUT2 puncta was normalized to the soma perimeter of each analyzed cell.

Images of VGLUT1 puncta were acquired using a 63X/1.4NA oil-immersion objective, with a scan zoom of 2 and a z-interval of 0.5 μm. Maximum intensity projections of sagittal sections near the apical edge of the ML were used for quantification. VGLUT1 puncta were identified using the Trainable Weka Segmentation (v4.0.0) plugin[123]. Both the number and size of VGLUT1 puncta were quantified, and VGLUT1 density was calculated as the number of puncta per ROI area.

Molecular layer interneurons (MLIs) stained with NeuroTrace and parvalbumin (PV) were imaged using a 20×/0.8NA objective with a z-interval of 1 μm. Maximum intensity projections of sagittal sections were analyzed using a macro implemented in FIJI. The number of NeuroTrace+, PV+, and colocalized NeuroTrace+ PV+ cells was counted. MLI density was calculated as the number of neuronal cells per ROI area.

Images of sagittal cerebellar sections were acquired using an Axio Imager.M2 epifluorescence microscope with a 10x/0.45 NA objective. Granule cell layer (GCL) thickness was measured at two distinct positions within each lobule: at the base and in the middle region. At each position, two independent measurements were taken, defined as the distance from the bottom of the Purkinje cell soma to the white matter. The mean value of these measurements was used for subsequent analysis. The area of each cerebellar lobule was determined by manually delineating lobule boundaries and measuring the enclosed area using FIJI.

### In utero electroporation
Timed-pregnant FVB/NHsd females at E12.5 were deeply anesthetized with isoflurane (5% induction, 2% maintenance) in oxygen at a flow rate of 1 l/min, and body temperature was maintained at 37 °C. Buprenorphine (0.1 mg/kg) was administered subcutaneously for analgesia. The abdominal cavity was then opened, and the uterus was carefully exposed to access the embryos. A beveled glass micropipette was used to inject a combination of 3 μg/μl of *pCAG-GFP* (Addgene; #11150; a gift from C. Cepko[124]) and 1 μg/μl of *pCAG-Kir2.1-T2A-tdTomato* (Addgene; #60598; a gift from M. Scanziani[125]) with 0.05% of FastGreen into the fourth ventricle of each embryo. Electroporation was performed by delivering five pulses of 35 V for 50 ms at 900 ms intervals using a 2 mm tweezertrode (CUY650P2, Nepa Gene) connected to an electroporator (BTX, ECM830). The electrodes were positioned to target Purkinje cell progenitors in the cerebellar ventricular zone. After electroporation, the uterus was placed back into the abdominal cavity, and the abdominal wall and skin were sutured. The female was monitored until complete recovery.

### Balance beam test
All behavioral tests were performed on adult mice (P60–P90).

Mice were tested on a 1-meter-long bean with a flat surface, 12 mm wide, elevated 50 cm above the surface between two platforms. A cage was placed at the end of the beam as a finish point. The test consisted of two training days followed by a testing day, with three trials per day. Trials in which a mouse fell five consecutive times were considered completed. Motor coordination was determined by averaging the time taken to cross the beam and the number of missteps per run[126].

### Accelerating the rotarod test
Mice were tested on an accelerating rotarod over five consecutive days, with four non-consecutive trials per day. On days 1 to 4, the rotating rod was accelerated to 40 rpm, and on day 5, the maximum speed was increased to 80 rpm. Mice were given an hour to rest between trials. A trial was considered complete when the mouse reached a maximum latency of 300 s, fell off the rod, or rotated three consecutive times. Motor coordination was quantified by averaging the latency to fall across trials on the accelerating rod (Ugo Basile Biological Research Apparatus).

### LocoMouse test
The LocoMouse test was performed to assess whole-body coordination during overground locomotion[30]. Mice crossed a glass corridor (66.5 cm long, 4.5 cm wide, and 20 cm high) between two dark boxes. A mirror (66 cm × 16 cm) was placed below the corridor at a 45° angle to allow simultaneous recording of side and bottom views. Side and bottom views of walking behavior were recorded using a high-speed camera (Basler, 1440 × 250 pixels, 400 fps). The test consisted of two consecutive habituation days followed by three consecutive experimental days. During habituation, mice walked freely between the two dark boxes for 15 min. On each experimental day, 15 trials per animal were collected; each trial consisted of a single corridor crossing, and the animals were not required to walk continuously throughout a trial. A DeepLabCut network was trained using the side camera view of the LocoMouse to track the paws, nose, and tail base positions of each individual mouse[127]. Data were processed using custom-written Python (v3.7) code available at GitHub (GitHub, San Francisco, CA, USA, https://github.com/BaduraLab/DLC_analysis)[31].

### Open field test
The open field test was performed in a white-opaque polypropylene square arena (50 × 50 × 40 cm) under uniform lighting. Each animal was placed in the center of the arena and allowed to explore freely for 10 min. A digital video system recorded the overall activity inside the box (Carl Zeiss Tessar 2.0/3.7 2MP V-UBM46 Autofocus Camera). Data were analyzed using OptiMouse, an open-source MATLAB program[128], which automatically calculated the total distance traveled and speed to evaluate spontaneous locomotor activity.

### Compensatory eye movements
Mice were surgically prepared for head-restrained recordings by placing a custom-built metal pedestal with a square magnet on the frontal and parietal bones under general anesthesia with isoflurane/O$_2$. Three days after recovery, mice were head-fixed in a holder at the center of a turntable (60 cm diameter), surrounded by a cylindrical screen (63 cm diameter) with a random-dotted pattern (drum). Eye movements were recorded with a CCD camera fixed to the turntable, and eye-tracking software was used (ETL-200, ISCAN Systems, Burlington, NA, USA). Eyes were illuminated with two table-fixed infrared emitters and a third emitter attached to the camera, which produced a tracked corneal reflection used as the reference point. The recording camera was calibrated by moving it left-right 20° peak-to-peak while the eye remained stationary[129]. Gain and phase values of eye movements were calculated using custom-made MATLAB scripts, available at GitHub (GitHub, San Francisco, CA, USA, https://github.com/MSchonewille/iMove;[130]). Mice were familiarized with the experimental setup three days before the experiments by providing visual and/or vestibular stimulations for 30 min. Compensatory eye movements were evoked using a sinusoidal drum rotation in light (OKR), turntable rotation in dark (vestibular ocular reflex-VOR), or turntable rotation in light (visual VOR-VVOR) with 5° amplitude at 0.1–1 Hz. To assess motor learning, mice were subjected to a mismatch between visual and vestibular input to adapt the VOR. The ability to perform VOR phase reversal was tested over five consecutive days, with six 5 min training sessions per day and VOR recordings before, between, and after each training session. Mice were kept in the dark between recording sessions to avoid unlearning of the adapted responses. On the first training day, the visual (drum) and vestibular (turntable) stimuli rotated in phase at 0.6 Hz, each with an amplitude of 5°, resulting in a decrease in gain. In the following days, the drum amplitude was increased to 7.5° (day 2) and 10° (days 3, 4,

and 5), while the turntable amplitude remained at 5°. This led to a reversal of the VOR direction, an inversion of the compensatory eye movement driven by vestibular input, moving the eye in the same direction as the head rotation rather than the normal compensatory opposite direction. Motor performance in response to these stimulations was measured by calculating the response gain (ratio of eye movement amplitude per stimulus amplitude) and phase (difference between eye and stimulus in degrees)[21].

## Eyeblink conditioning

A custom-built metal pedestal with a square magnet on top was placed on the frontal and parietal bones of a mouse under general anesthesia with 2% isoflurane/$O_2$. Three days after recovery, mice were head-fixed on a foam-cylindrical treadmill, on which they could walk freely inside custom-built sound- and light-attenuating boxes. Eyelid movements were monitored under infrared illumination using a high-speed (333 fps) monochrome Basler video camera (Basler ace 750–30 gm). Stimuli and measurement devices were controlled by National Instruments hardware and custom-written LabVIEW software. The conditioned stimulus (CS) was a blue LED light (duration 280 ms, diameter 5 mm) placed 10 cm in front of the mouse. The unconditioned stimulus (US) consisted of a 30 ms mild corneal air puff, controlled by a VHS P/P solenoid valve with a back pressure of 30 psi (Lohm rate, 4750 Lohms; Internal volume, 30 µl, The Lee Company®, Westbrook, USA) and delivered via a 27.5 mm gauge needle perpendicularly positioned at approximately 5 mm from the center of the left cornea. Prior to the experiments, mice were habituated to the set-up for one day (two 30 min sessions spanning 6 h) with no stimuli delivered. To assess motor learning, mice were subjected to EBC training for 5 consecutive days, with two sessions per day separated by 6 h of rest. Before the first training session, a baseline session was conducted to confirm that the CS did not elicit any reflexive eyelid closure. This baseline session consisted of 15 CS-only trials and 3 US-only trials. Immediately after this, the training sessions began. During each session, every animal received a total of 200 paired CS-US trials, 20 US-only trials, and 20 CS-only trials. These trials were presented across 20 blocks, with each block consisting of 1 US-only trial, 10 paired CS-US trials, and 1 CS-only trial. The interval between CS onset and US onset was 250 ms. All experiments were performed at approximately the same time of the day by the same experimenter. Individual eyeblink traces were analyzed with a custom-written MATLAB script available at GitHub (GitHub, San Francisco, CA, USA, https://github.com/francescafiocchi91/Eyeblink_Conditioning[36]) (R2018a, Mathworks). Eyeblink traces (2000 ms) were imported from a MySQL database into MATLAB and aligned at zero for the 500 ms pre-CS baselines. Trials with significant activity in the 500 ms pre-CS period (>7 times the interquartile range) were considered invalid and disregarded for further analysis. The eyelid signal was min–max normalized so that a fully open eye corresponded to a value of 0 and a fully closed eye to a value of 1[131]. In valid normalized CS-only trials, eyelid responses were considered as conditioned responses (CR) if the maximum amplitude was larger than 0.10 between 100 and 500 ms after CS onset and showed a positive slope in the 150 ms before the time point of the expected US delivery (US is omitted in CS-only trials)[132].

## Purkinje cell collection for RNA sequencing

To isolate Purkinje cells, the cerebellar tissue was extracted from either P7 *Pcp2^+/+;Kir2.1* or P7 *Pcp2^cre/+;Kir2.1* mice. Sagittal slices of vermal cerebellar tissue were obtained and maintained as described in the ex vivo Purkinje cell recordings section. For cell collection, individual slices were transferred to a recording chamber and maintained at 34 °C under continuous perfusion with equilibrated aCSF. In P7 *Pcp2^cre/+;Kir2.1* slices, Purkinje cells were selected based on Kir2.1-mCherry expression. Neurons were extracted into a glass pipette using negative pressure, and 50 cells were sampled across the cerebellar cortex. Eight biological samples per genotype were collected for RNA sequencing. Cells were harvested in a mild hypotonic lysis buffer containing 0.2% Triton X and 2 U/µl RNase inhibitor, kept on ice during collection, and either processed immediately or stored at -80 °C until RNA extraction.

## cDNA library preparation and RNA sequencing

Library preparation and RNA sequencing experiments were performed at the Erasmus Center for Biomics (Erasmus MC, Rotterdam, The Netherlands). For each biological replicate, cDNA libraries were generated using the Smart-seq2 method[133]. Briefly, cells were lysed, and poly(A) + RNA was reverse-transcribed using an oligo(dT) primer. Template switching during reverse transcription was achieved using a locked nucleic acid-containing template-switching oligonucleotide. The reverse-transcribed cDNA was preamplified with primers for 18 cycles, followed by cleanup. Tagmentation was performed on 1 ng of the pre-amplified cDNA using the Illumina Nextera DNA Flex kit. The tagmented library was extended with Illumina adapter sequences by PCR for 12 cycles and purified. The resulting sequencing library was measured on a Bioanalyzer and equimolarly loaded onto a flow cell for sequencing according to the Illumina TruSeq v3 protocol on the Illumina HiSeq 2500 platform, with a single read of 50 base pairs and dual 9-base-pair indices.

## Bioinformatic analysis of RNA sequencing

Illumina adapter sequences and poly-A stretches were trimmed from the reads prior to mapping to the mouse reference sequence GRCm38 (mm10) using HISAT2 (version 2.1.0). From the alignments, the reads per gene were determined using htseq-count (version 0.11.2). Gene and transcript annotations were retrieved from Ensembl (build 101) using consensus coding sequences for each gene. Count data were analyzed using RStudio (version 0.99.484) following the edgeR-limma package workflow[134]. Briefly, Ensembl IDs were annotated to Entrez gene IDs, gene symbols, gene names, and chromosome locations using the Mus.musculus package. Raw counts were transformed into counts per million (CPM) and $\log_2$-CPM, and genes with a CPM below 0.2 in at least three samples were filtered out. Normalization of gene expression distributions was performed using the Trimmed Mean of M-values method.

DEGs between Purkinje cells from *Pcp2^+/+;Kir2.1* and *Pcp2^cre/+; Kir2.1* mice were visualized using a hierarchical clustering heatmap with normalized row z-scores in R. To identify biologically significant genes, data were organized in a volcano plot with a fold change cutoff of 0.5 and a $-\log_{10}$(false discovery rate; FDR) cutoff of 1.3. Gene ontology (GO) analyses were performed using the Gene Ontology tool (http://pantherdb.org/), and KEGG enrichment analyses were performed using the clusterProfiler package (version 4.14.6)[135]. DEGs were selected for GO and KEGG enrichment analysis. The complete gene set identified from the biological samples was used as the reference list, and GO terms with corrected $P \le 0.05$ were considered significantly enriched. GO term visualization was generated using the ggplot2 package (version 3.5.1), and network visualizations using the igraph package (version 2.1.4)[136]. A list of 216 human movement disorder-associated genes was extracted from the KEGG Disease database (https://www.genome.jp/kegg/disease/), which compiles known genetic and environmental contributors to human diseases. This list was compared to the 1602 DEGs identified from RNA sequencing, revealing 25 overlapping genes. The shared genes were ranked by $-\log_{10}$(FDR) and $\log_2$ fold change, *Prkcg* and *Car8* emerging as the top hits.

## Single-molecule fluorescence in situ hybridization

Mice were perfused as described above, and brains were postfixed for 2 h at 4 °C before being cryoprotected in 30% sucrose/0.1 M PB

overnight at 4 °C. Sagittal sections 30 μm thick were obtained using a freezing microtome, mounted on RNAse-free SuperFrost Plus slides (Epredia), and probed against the candidate genes according to the manufacturer's protocol. Single-molecule fluorescence in situ hybridization was performed using the RNAscope Multiplex Fluorescent Reagent Kit v2 (ACDBio, #323100) with probes targeting *Car8* (*Mm-Car8-C2*, #514171) and *Prkcg* (*Mm-Prkcg-C2*, #417911). Sections were co-immunostained with rabbit anti-RFP (1:1000, Rockland, #600-401-379) and Cy3-AffiniPure Donkey anti-Rabbit (1:1000, Jackson ImmunoResearch, #711-165-152).

## Cell culture and transfection

HEK293T cells were cultured in Dulbecco's Modified Eagle's medium supplemented with high glucose, Glutamax, and pyruvate (DMEM; ThermoFisher Scientific, #31966-047). In addition, 10% fetal bovine serum, 1% of penicillin (10,000 U/ml), and of streptomycin (10,000 μg/ml). Cultures were maintained at 37 °C in a humidified atmosphere with 5% $CO_2$. HEK293T cells were plated in 6-well plates at a density of $0.5 \times 10^6$ cells/well and transfected the following day with plasmids expressing the full-length genes of interest and respective shRNA-expressing plasmids at a 1:3 ratio using Lipofectamine 2000 (ThermoFisher Scientific, #11668019). The plasmids used to express *Prkcg* and *Car8* were *pcDNA3.1+/C-(K)DYK-Prkcg* (GenScript, OMu17867) and *pcDNA3.1+/C-(K)DYK-Car8* (GenScript, OMu00430), respectively. shRNA sequences targeting *Prkcg* and *Car8* were designed by The RNAi Consortium (TRC) and obtained from Horizon Discovery. The antisense sequence and respective clone IDs were as follows: *Prkcg shRNA1* 5′-TTG ATG GCA TAG AGT TCG TCG-3′ (TRCN0000022684), *Prkcg shRNA2* 5′-TAA TAT GGA TCT CAT CCG ACG-3′ (TRCN0000022685), *Prkcg shRNA3* 5′-ATA GGC AAT GAT CTC AGG TGC-3′ (TRCN0000022686), *Prkcg shRNA4* 5′-TTC ACA TAA GTA AAG CCC TGG-3′ (TRCN0000022687), *Prkcg shRNA5* 5′-AAA CGA GCG GTG AAC TTG TGG-3′ (TRCN0000022688), *Car8 shRNA1* 5′-AAT ATC CAG GTA ACT CCT TCG-3′ (TRCN0000114512), *Car8 shRNA2* 5′-TTT AGG TTG ATA GGT GAC TGG-3′ (TRCN0000114513), *Car8 shRNA3* 5′-TAG CAT CAG GAA ACA CTA AGC-3′ (TRCN0000114514) and *Car8 shRNA4* 5′-ATC TGG GAT ATA GTT AAA GGG-3′ (TRCN0000114515). A non-targeting control *shRNA* against *lacZ* 5′-AAA TCG CTG ATT TGT GTA GTC-3′ was used as a negative control.

## Western blotting

HEK293T cells were collected 24 h after transfection in lysis buffer containing 50 mM Tris-HCl pH 8, 150 mM NaCl, 1% Triton X-100, 0.5% sodium deoxycholate, 0.1% SDS, and protease inhibitor cocktail. Protein concentrations were measured using the Pierce BCA Protein Assay Kit (Thermo Fisher Scientific). Samples were denatured, and equal amounts of proteins were separated by SDS-PAGE in Criterion TGX Stain-Free Gels (Bio-Rad) and transferred onto nitrocellulose membranes using the Trans-Blot Turbo Blotting System (Bio-Rad). Membranes were blocked with 5% bovine serum albumin (BSA; Sigma-Aldrich) in Tris-buffered saline (TBS)-Tween (20 mM Tris-HCl pH 7.5, 150 mM NaCl, and 0.1%, Tween 20) for 1 h at room temperature and incubated overnight at 4 °C with the following primary antibodies: rabbit anti-DYKDDDDK-tag (1:2000, GenScript, #A00170) and mouse anti-actin (1:1000, Millipore, #MAB1501). After washing, membranes were incubated for 1 h at room temperature with the following secondary antibodies: goat anti-Rabbit Immunoglobulins/HRP (1:10000, Agilent Dako, #P0448) or goat anti-Mouse Immunoglobulins/HRP (1:10000, Agilent Dako, #P0447). Protein bands were detected by the luminol-based enhanced chemiluminescence method (SuperSignal West Femto Maximum Sensitivity Substrate or SuperSignal West Dura Extended Duration Substrate, Thermo Fisher Scientific). Membranes were stripped with Restore PLUS Western Blot Stripping Buffer (Thermo Fisher Scientific) when necessary. Densitometry analyses of protein bands were normalized to actin levels using the Image Studio Lite software (LI-COR Biosciences).

## Generation of DNA constructs

DNA vectors were generated using standard molecular biology procedures. To produce viral vectors expressing *shRNA* for the target genes, pairs of oligonucleotides were designed, obtained (Integrated DNA Technologies) and hybridized for *shlacZ* (strand 1: 5′-cta ggA AAT CGC TGA TTT GTG TAG TCT GAT ATG TGC AGA CTA CAC AAA TCA GCG ATT TTT TTTg-3′and strand 2: 5′-aat tca aaaa AAA TCG CTG ATT TGT GTA GTC TGC ACA TAT CAG ACT ACA CAA ATC AGC GAT TTc-3′), *shPrkcg* (strand 1 5′-cta ggT AAT ATG GAT CTC ATC CGA CGC TCG AGC GTC GGA TGA GAT CCA TAT TAT TTTTg-3′and strand 2 5′-aat tca aaaa TAA TAT GGA TCT CAT CCG ACG CTC GAG CGT CGG ATG AGA TCC ATA TTAc-3′) and *shCar8* (strand 1 5′-cta ggT TTA GGT TGA TAG GTG ACT GGC TCG AGC CAG TCA CCT ATC AAC CTA AAT TTTTg-3′and strand 2 5′-aat tca aaaa TTT AGG TTG ATA GGT GAC TGG CTC GAG CCA GTC ACC TAT CAA CCT AAAc-3′). Each double-stranded oligonucleotide was subsequently cloned into the *pDIO-DSE-mCherry-PSE-MCS* vector (Addgene; #129669; a gift from B. Rico[137]) using EcoRI and AvrII restriction sites.

## Generation of viral particles

The adenovirus-associated virus (AAV) preparation AAV-FLEX-GFP was commercially acquired (Addgene; #28304-PHPeB; a gift from E. Boyden). HEK293T cells cultured in DMEM (with 10% fetal calf serum and penicillin/streptomycin) were co-transfected with AAV2/9 serotype helper plasmid, pAdΔF6 helper plasmid, and the pAAV-shRNA construct using polyethylenimine (PEI; 25 kDa, linear). Seventy-two hours after transfection, cells were harvested, lysed, and centrifuged to remove cellular debris. AAV particles were purified from the supernatant using an iodixanol density gradient and subsequently concentrated in 5% sucrose/PBS using an Amicon Ultra-15 centrifugal filter. Viral titers were determined by quantitative PCR targeting the *mCherry* reporter sequence present in the viral genome (vg). *The mCherry* primers used were: 5′-tcc cac aac gag gac tac ac-3′ and 5′-ctt gta cag ctc gtc cat gc-3′. The final viral titers ranged from $5 \times 10^{12}$ to $2 \times 10^{13}$ vg/ml[138].

## Intraventricular viral injection in neonates

For intraventricular injections, P0 *Pcp2^{cre/+}* were cryo-anesthetized with ice for 1 min before injection. Pups were placed in a custom-made platform, and a solution of AAVs diluted 1:100 in sterile NaCl containing 0.05% FastGreen was injected bilaterally into the lateral ventricles using a Nanoject III (Drummond Scientific Company). The injection site was located at approximately two-fifths of the distance between the lambda and each eye, and 1 μl of viral solution was injected into each lateral ventricle. After injection, pups were placed on a 37 °C warming pad to recover from anesthesia, and subsequently returned to the mother[139].

## Statistical analysis

Statistical analyses were performed using GraphPad Prism version 8.0.0 (GraphPad Software, San Diego, CA, USA; www.graphpad.com), MATLAB (MathWorks, Natick, MA, USA), and R software. Data normality was assessed using the Shapiro–Wilk test, and equality of variances was tested using the *F*-test. Statistical comparison between two normally distributed groups was performed using a two-tailed unpaired Student's *t*-test, or a two-tailed Mann–Whitney *U*-test for non-parametric data. For comparisons among more than two groups, one-way or two-way ANOVA was used for normally distributed data, while the Kruskal–Wallis test or a mixed-effects model was applied for non-parametric data. Statistical significance was defined as $P < 0.05$.

**Reporting summary**

Further information on research design is available in the Nature Portfolio Reporting Summary linked to this article.

## Data availability

Sequencing data generated in this study have been deposited in the National Center for Biotechnology Information (NCBI) Gene Expression Omnibus (GEO) under the accession number GSE294208 and are publicly available. Source data are provided with this paper.

## Code availability

This study did not generate original codes. For LocoMouse, Compensatory Eye Movements, and EBC analyses, the codes we previously reported and deposited on GitHub: https://github.com/BaduraLab/DLC_analysis[31], https://github.com/MSchonewille/iMove[130], and https://github.com/francescafiocchi91/Eyeblink_Conditioning[36], respectively.

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

## Acknowledgements

The authors kindly thank all personnel of the animal facility at Erasmus Medical Center Rotterdam, as well as Elize Haasdijk, Erika Sabel-Goedknegt, and Doriane Gisquet for excellent technical assistance. We also thank Guillermina López-Bendito for providing the Kir2.1 (Gt(ROSA) 26Sor$^{tm2(CAG-KCNJ2/mCherry)Fmr}$) mouse model. This work was supported by H2020 European Research Council (ERC Starting Grant #680235), Erasmus MC Flagship Convergence Incentive Grant, and Dutch Research Council (NWO) through OCENW.M.22.046 to M.S.; Netherlands Organization for Health Research and Development (ZonMW Off-road grant 451001027), NWO through OCENW.XS5.121 and Incentive Grant for Women in STEM (19498) to C.O.; NWO VENI fellowship (016.Veni.192.270), and OCENW.XS21.1.087 to J.J.W.; China Scholarship Council (202407720046) to J.Y.; and Royal Netherlands Academy of Arts and Sciences (KNAW) Research Fund and the Leiden Regenerative Medicine Platform for AAV vector production to F.d.W.

## Author contributions

Conceptualization—C.O., L.L., G.M.v.W., and M.S. Methodology—C.O., J.J.W., and M.S. Investigation—C.O., J.J.W., P.T.-S., N.M., F.O., F.K., L.P., E.R., and M.S. Data curation—C.O., J.J.W., P.T.-S., J.Y., N.M., F.O., F.K., Z.H., I.S., S.T., and M.S. Formal analysis—C.O., J.J.W., P.T.-S., J.Y., F.R.F., I.S., and M.S. Resources—C.O., J.J.W., F.d.W., F.R.F., S.T., Z.O., M.C.G.N.H., W.F.J.I., G.L.-B., A.B., and M.S. Software—C.O., J.J.W., J.Y., F.R.F., S.T. and M.S. Visualization—C.O. Writing—original draft—C.O. Writing—review and editing—all authors. Funding acquisition—C.O., J.J.W., J.Y., F.d.W., and M.S. Supervision—C.O., J.J.W., and M.S.

## Competing interests

The authors declare no competing interests.
