## [Transparent Peer Review file · Nature Communications]

Purkinje cell intrinsic activity shapes cerebellar development and function

Corresponding Author: Dr Catarina Osório

Version 0:

Reviewer comments:

Reviewer #1

(Remarks to the Author)

In this manuscript, the authors aimed to study how a Purkinje cell-specific reduction of intrinsic activity, achieved through the overexpression of the inward rectifier Kir2.1 channel, affects Purkinje cell (PC) development and cerebellar-dependent behaviors in adult mice. Using a conditional *Pcp2creERT2* strain of mice, they demonstrated that a single dose of tamoxifen administered at P1 led to PC hyperpolarization and morphological alterations in PC dendrites. In non-conditional *Pcp2cre;Kir2.1* mice, they identified synaptic alterations at PC-DCN contacts and showed that adult mice exhibited impairments in specific cerebellar learning tasks, such as VOR adaptation and eyeblink conditioning. The authors also examined how Kir2.1 expression at specific time points during postnatal development influenced dendritic morphology after 7 days and affected motor balance in adulthood. Finally, they compared gene expression profiles between non-conditional *Pcp2cre;Kir2.1* mice and WT mice at P7, identifying two differentially expressed genes, *Prkcg* and *car8*. By targeting them using ShRNA at birth they showed morphological alterations at P7

Overall, this is a potentially interesting study that aggregates a collection of well-performed and analyzed experiments. While this is not the first study to investigate the effects of Purkinje cell (PC) hyperpolarization (e.g., see PMID: 19805323), this manuscript provides convincing evidence that PC hyperpolarization during development, particularly within the first two postnatal weeks, leads to morphological, functional, and behavioral impairments. However, several flaws significantly limit the impact of this manuscript in its current form.

Major Concerns:

1-The study lacks controls to confirm that: (1) The different strains of *Pcp2* mice express Kir2.1 in similar subsets of PCs. (2) Different tamoxifen injection protocols (single dose, low dose, high dose, or injections at different ages) result in interpretable and consistent levels of Kir2.1 expression.

2- The study is essentially descriptive with no attempt to assess mechanisms involved in the dysfunctions observed. Even the RNA sequencing analysis falls short in identifying putative mechanisms.

Specific points:

- Expression of Kir2.1 across mice strains: line 73 suggests that not all Purkinje cells are targeted in *Pcp2creERT2;Kir2.1* mice. At a minimum, a thorough evaluation of transgene expression in both strains of mice is necessary to understand its potential influence on cerebellar development and behavior.

- Functional assessment of PC activity: it is critical to evaluate Purkinje cell activity *in vivo*, as well as the output activity of the cerebellum in the deep cerebellar nuclei (DCN).

- Integration of parallel fiber and climbing fiber inputs: understanding how parallel fiber and climbing fiber inputs are integrated by Purkinje cells is essential to interpret the behavioral outcomes. For example, see PMID: 19805323.

Additionally, assessing climbing fiber multi-innervation (e.g., through acute slice experiments) would be informative. At the very least, VGluT2 staining to evaluate climbing fiber inputs is missing.

- PC-DCN immunostaining: immunostaining in non-conditional mice or mice injected with high doses of tamoxifen provides limited information, as co-localization staining lacks functional relevance. Functional data, such as electrophysiological recordings from brain slices, would be far more informative. Furthermore, the switch to high-dose tamoxifen when studying PC-DCN connections during development is abrupt and unexplained. Was the single-dose protocol used in Figure 1 ineffective?

- Non-conditional expression extrapolations: the activity of PCs in *Pcp2cre;Kir2.1* (non-conditional) mice was not tested, yet conclusions are extrapolated from the conditional model injected with tamoxifen at P1. Similarly, VOR and eyeblink conditioning experiments were only performed in the non-conditional model. This discrepancy should be addressed.

- A diagram summarizing the experimental design, including the different strains of mice, tamoxifen doses, and the resulting

effects, is necessary to improve clarity and guide readers through the manuscript.

(Remarks on code availability)

-

Reviewer #2

(Remarks to the Author)

This is an elegant and well-written manuscript that tries to demonstrate the role that Purkinje cell activity plays in the development of the cerebellum. Using an inducible transgenic gene approach, the authors explore the temporal properties of intrinsic activity of Purkinje cells in the proper formation of cerebellar circuits.

The figures are clear and the experiments are for the most part quite thorough. The authors try to reduce Purkinje cell activity both early during development and later using inducible expression of Kir2.1 at different ages. Thus, the authors try to establish that its role is most important during early development, with the strongest effects on Purkinje cell dendritic length and behavior being observed when activity is impaired at the earliest postnatal ages.

Overall I think that this manuscript reasonably demonstrates the importance of Purkinje cell activity in Purkinje cell dendritic growth and in cerebellar function, although more evidence about exactly what is happening with firing would help the manuscript (see below). If these changes are incorporated, I believe that it will be a classic paper that will be well cited both in the cerebellar field and in the field of circuit development, meaning its impact will be large and thus is suitable for publication in Nature Comms.

While I think that the manuscript is very interesting, I am not yet fully convince by the data that expressing Kir2.1 is doing so via Purkinje cell activity and not some other means (such as making the cells sick). Points 1 and 2 would clarify this and provide additional strength to their hypothesis.

1. The firing data presented in Extended Data Figure 1d-e shows that the Purkinje cells expressing Kir2.1 at 21 days old can still spike, although it could be highlighted better by showing current injection steps for additional representative traces to better demonstrate the right-ward shift, which is unfortunately not particular evidence with the 3 traces shown for each example. It would be beneficial to see traces with fewer action potentials so a better comparison of individual action potentials could be observed as well. I am surprised that there is no activity at 0 pA for the two control conditions given that Purkinje cells typically fire spontaneously at P21, and would like the authors to comment on this. Also, the magnitude of the voltage deflection to the negative current injection is much smaller in the Kir2.1 expressing cells, likely because the cells are smaller. I think this could be supported by reporting the whole cell capacitance and input resistance (Rin) for their recordings. I think this data should be incorporated into Fig. 1 as without that, the impression arising from Figure 1 is that firing is completely abolished, which is likely incorrect (and also brings up the question of whether the cells can fire action potentials at all).

2. How Kir2.1 affects Purkinje cell activity at young ages is demonstrated in Extended Figure 4. However, the authors do not focus on activity, but rather report membrane potential and the amplitude of a current evoked at a negative holding potential. This seems counter-intuitive to me, given that we know that Purkinje cells fire action potentials at these young ages. Firing data would be much more convincing and informative in my opinion, since firing is what they are trying to manipulate. I suggest that this is done as described above in point 2. The authors argue that the effects of Kir2.1 are mediated through the alteration in firing activity, so I think that needs to be shown directly. Without direct evidence of firing differences, it could be that Kir2.1 is having an effect by making the cells sicker or decreasing their Rin and making them leakier, rather than directly through firing.

3. The manuscript focuses heavily on Purkinje cell structure, and relate this to behavior. Yet Purkinje cells are known to influence the development of other parts of the cerebellar circuit. It would be useful if the authors could determine if other parts of the circuit are changed by this manipulation of Purkinje cell activity. While I do not think it is required to characterize everything, at least the size of the lobules and the thickness of the granule cell layer (as a read-out of granule cell number), would be informative. The MLI number would also be interesting, since there are dendritic changes in Purkinje cells. This would help understand the extent of the changes in the circuit and whether behavioral changes are likely to arise entirely from Purkinje-cell related changes or may also involve other parts of the circuit. I think this is most important for the P1 and P7 injections since this is when granule cells and MLIs are still migrating.

4. Finally, the authors state that they label only a subset of Purkinje cells with these methods. It would be beneficial to understand approximately how many Purkinje cells are labeled and in which lobules they tend to see labeling, if indeed there is some regionalization of expression of their Kir2.1 transgene.

(Remarks on code availability)

n/a

Version 1:

Reviewer comments:

Reviewer #1

(Remarks to the Author)

The authors have convincingly addressed my main concerns. They performed additional experiments and provided extensive analyses that clarify ambiguities present in the previous version of the manuscript. The revised manuscript is also easier to read and more clearly demonstrates the importance of Purkinje cell intrinsic excitability in cerebellar development. Overall, this work represents one of the most comprehensive datasets on this topic compared with prior studies.

Nonetheless, the following related study should also be cited:

Lorenzetto E, Caselli L, Feng G, Yuan W, Nerbonne JM, Sanes JR, Buffelli M. *Genetic perturbation of postsynaptic activity regulates synapse elimination in developing cerebellum.* **Proc Natl Acad Sci U S A.** 2009 Sep 22;106(38):16475–80.

My remaining concern is that the main Figure 5 should be extended to include the data currently shown in Supplementary Figure 10 (panels I–L). In its current form, Figure 5 does not clearly convey that phenotype severity depends on the timing of induction. Adding these panels would strengthen the figure and support this key conclusion.

Reviewer #2

(Remarks to the Author)

I am happy with the response to my questions, and feel that the changes incorporated in the revised manuscript significantly strengthen it. I have no further concerns with the manuscript. I think it will add to our understanding of cerebellar development, an important topic.

Response to reviewers for Nature Communications manuscript NCOMMS-24-63742-T by Osório et al.

We sincerely thank the reviewers for their insightful and constructive comments. Their feedback has been invaluable in improving the clarity, rigor, and overall quality of our manuscript.

Reviewer #1 (Remarks to the Author):

In this manuscript, the authors aimed to study how a Purkinje cell-specific reduction of intrinsic activity, achieved through the overexpression of the inward rectifier Kir2.1 channel, affects Purkinje cell (PC) development and cerebellar-dependent behaviors in adult mice. Using a conditional Pcp2creERT2 strain of mice, they demonstrated that a single dose of tamoxifen administered at P1 led to PC hyperpolarization and morphological alterations in PC dendrites. In non-conditional Pcp2cre;Kir2.1 mice, they identified synaptic alterations at PC-DCN contacts and showed that adult mice exhibited impairments in specific cerebellar learning tasks, such as VOR adaptation and eyeblink conditioning. The authors also examined how Kir2.1 expression at specific time points during postnatal development influenced dendritic morphology after 7 days and affected motor balance in adulthood. Finally, they compared gene expression profiles between non-conditional Pcp2cre;Kir2.1 mice and WT mice at P7, identifying two differentially expressed genes, Prkcg and car8. By targeting them using ShRNA at birth they showed morphological alterations at P7.

Overall, this is a potentially interesting study that aggregates a collection of well-performed and analyzed experiments. While this is not the first study to investigate the effects of Purkinje cell (PC) hyperpolarization (e.g., see PMID: 19805323), this manuscript provides convincing evidence that PC hyperpolarization during development, particularly within the first two postnatal weeks, leads to morphological, functional, and behavioral impairments. However, several flaws significantly limit the impact of this manuscript in its current form.

Response: We thank the reviewer for their overall appreciation of our work and their constructive feedback.

Major Concerns:

1-The study lacks controls to confirm that: (1) The different strains of *Pcp2* mice express *Kir2.1* in similar subsets of PCs. (2) Different tamoxifen injection protocols (single dose, low dose, high dose, or injections at different ages) result in interpretable and consistent levels of *Kir2.1* expression.

Response: We thank the reviewer for the recommendation to include additional controls addressing Purkinje cell labeling across mouse strains and tamoxifen protocols at different ages. Tamoxifen dosing and timing were adjusted to achieve either sparse labeling or labeling in all Purkinje cells at different developmental ages, enabling controlled expression of tdTomato or *Kir2.1*-mCherry in *Pcp2^{creER};Ai14* and *Pcp2^{creER};Kir2.1*, respectively. To ensure recombination of all Purkinje cells from birth while avoiding multiple injections in neonatal pups, we also used constitutive *Pcp2^{cre/+};Kir2.1* mice and their control littermates. This approach ensured robust expression of tdTomato or *Kir2.1*-mCherry in all Purkinje cells from birth onward. An overview of all experimental designs is now provided in **Supplementary Fig. 14**. We have included the requested controls in the revised manuscript, specifically:

Supplementary Fig. 1a-d Sparse labeling of Purkinje cells at P21 in *Pcp2^{creER};Ai14* and *Pcp2^{creER};Kir2.1*;

Supplementary Fig. 7 Sparse labeling of Purkinje cells at P7, P14, or P21 in *Pcp2^{creER};Ai14* and *Pcp2^{creER};Kir2.1*;

Supplementary Fig. 1e-g Labeling of all Purkinje cells at P21 in *Pcp2^{+/+};Kir2.1* and *Pcp2^{cre/+};Kir2.1*;

Supplementary Fig. 9 Labeling of all Purkinje cells at P7 in *Pcp2^{+/+};Kir2.1* and *Pcp2^{cre/+};Kir2.1* and at P14 or P21 in *Pcp2^{+/+};Kir2.1* and *Pcp2^{creER};Kir2.1*;

Supplementary Fig. 10 Labeling of all Purkinje cells in the adult in *Pcp2^{+/+};Kir2.1* and *Pcp2^{cre/+};Kir2.1*, and *Pcp2^{+/+};Kir2.1* and *Pcp2^{creER};Kir2.1*.

These additions clarify that labeling patterns are consistent across mouse strains and tamoxifen induction protocols. The corresponding text has been incorporated into the **Methods** section (pages 20-21, lines 475-488) and **Supplementary Information** of the revised manuscript.

2- The study is essentially descriptive with no attempt to assess mechanisms involved in the dysfunctions observed. Even the RNA sequencing analysis falls short in identifying putative mechanisms.

Response: We appreciate the reviewer's valuable comment. Although our initial Discussion addressed potential mechanisms underlying the observed dysfunctions, we appreciate the opportunity to expand and clarify this important aspect of our study. To address the reviewer's suggestion, we have extended our RNA sequencing analysis to more comprehensively identify molecular pathways and gene networks contributing to the observed phenotypes. These analyses are now presented in new **Figure 6** (page 53-54, lines 1286-1298) and **Supplementary Fig. 12**. We have clarified the specific purpose of each analysis and more explicitly linked the transcriptional changes to the morphological, functional, and behavioral phenotypes observed. This includes highlighting pathways related to neuronal maturation, synaptic signaling, ion channel regulation, and cerebellar circuit development that may contribute to the dysfunctions observed upon Kir2.1 expression. Please find these new changes in the revised manuscript in the **Methods** (pages 31-32, lines 767-790), **Results** (pages 12-13, lines 252-283), and **Discussion** (pages 15-19, lines 336-340, 349-360, 417-422, 447-455) sections. We hope the reviewer finds that these additions strengthen the mechanistic insights and overall impact of our study.

Specific points:

- Expression of Kir2.1 across mice strains: line 73 suggests that not all Purkinje cells are targeted in Pcp2creERT2;Kir2.1 mice. At a minimum, a thorough evaluation of transgene expression in both strains of mice is necessary to understand its potential influence on cerebellar development and behavior.

Response: We thank the reviewer for highlighting the need to further clarify Kir2.1 expression across mouse strains. The goal of the experiments shown in new **Figure 2a-g** and **Figure 4** was to investigate whether disrupting intrinsic activity impacted Purkinje cell morphological development. To enable high-resolution dendritic reconstruction in a densely packed brain structure, we deliberately employed a sparse labeling strategy using the *Pcp2^{creER}* line with a single tamoxifen injection (100mg/kg) at postnatal day 1. This approach generated sparse

tdTomato or Kir2.1-mCherry expression in Purkinje cells, preventing overlap with neighboring cells and thereby allowing reliable quantification of individual dendritic arbors. It is important to note that in this model, Kir2.1 expression in Purkinje cells following tamoxifen injection is an all-or-none event, as Cre-mediated recombination removes the STOP cassette, allowing Kir2.1 expression (now shown in the new **Supplementary Fig. 1**). Thus, once recombined, Kir2.1 expression is consistent within each labeled Purkinje cell. Furthermore, both *Ai14* and *Kir2.1* mouse lines were maintained as homozygous and crossed with Cre-driver lines, ensuring that all progeny were heterozygous for *Ai14* and *Kir2.1* and therefore maintained consistent transgene expression levels. To further validate the robustness of our findings, we obtained comparable results using a different approach for sparse labeling, i.e. using *in utero* electroporation, as shown in new **Supplementary Fig. 3**. While the promoter driving Kir2.1 is the same between approaches - the CAG promoter in the *in utero* electroporation experiments and a *CAG-lox-STOP-lox-Kir2.1-mCherry* cassette inserted into the Rosa26 locus in the mouse model - we cannot directly compare Kir2.1 expression levels across these systems. Nevertheless, the convergent results strongly suggest that small differences in Kir2.1 expression levels do not materially affect the observed phenotype. Representative images of sparse labeling are now included in the revised manuscript in **Supplementary Fig. 1a-d** and **Supplementary Fig. 7**.

For all other experiments, *in vivo* extracellular recordings, synaptic terminal quantification, behavioral tests, and RNA sequencing, we employed a genetic strategy to achieve labeling of all Purkinje cells. This was accomplished either by modifying tamoxifen dosing and injection frequency in the inducible *Pcp2^{creER};Ai14* and *Pcp2^{creER};Kir2.1* mice, or by using the non-inducible *Pcp2^{cre/+};Kir2.1* mouse model. Corresponding images of these protocols have been added to **Supplementary Fig. 1e-g**, **Supplementary Fig. 9**, and **Supplementary Fig. 10**.

For experiments requiring recombination from birth, we used non-inducible *Pcp2^{cre/+};Kir2.1* mice and their littermate controls, while avoiding multiple injections in neonatal pups. This ensured Kir2.1-mCherry expression in all Purkinje cells from the earliest developmental stages. Previous studies detected *Pcp2* mRNA at P1¹ and GFP expression at E17 in a *Pcp2-GFP* transgenic mice². We found that in *Pcp2^{cre/+}(B6.Cg-Tg(Pcp2-cre)3555Jdhu/J)*³ we could detect tdTomato or Kir2.1-mCherry at P1 in *Pcp2^{cre/+};Ai14* and *Pcp2^{cre/+};Kir2.1* mice, respectively (**Reviewers Fig. 1** below). At P1, it is possible to distinguish parasagittal bands of tdTomato (**Reviewers Fig. 1c**) and Kir2.1-mCherry (**Reviewers Fig. 1c**) expression characteristic of the *Pcp2* gene at this age. By P7, *Pcp2^{cre/+};Kir2.1* mice show labeling in all Purkinje cells, but with a higher intensity in the nodular regions of the cerebellum due to the *Pcp2* gene expression pattern (this gene is first expressed

in parasagittal bands overlapping Zebrin-positive cells, and then the expression extends to all Purkinje cells) (new **Supplementary Fig. 9a, b**).

Reviewers Fig. 1 Specific labeling of Purkinje cells at P1 in *Pcp2*^{cre/+};Ai14 and *Pcp2*^{cre/+};Kir2.1

a Experimental design. Brain from conditional *Pcp2*^{cre/+};Ai14 was collected at P1. **b** Schematic illustrating the genetic strategy used to express tdTomato specifically in Purkinje cells, showing that upon Cre-mediated recombination, it is an all-or-none event. **c** Coronal cerebellar sections at P1 from *Pcp2*^{cre/+};Ai14 mice show labeling of tdTomato (magenta) and DAPI (grey). **d** Experimental design. Brain from conditional *Pcp2*^{cre/+}; Kir2.1 mice was collected at P1. **e** Schematic illustrating the genetic strategy used to express Kir2.1-mCherry specifically in Purkinje cells. **f** Coronal cerebellar sections at P1 from *Pcp2*^{cre/+};Kir2.1 mice show labeling of Kir2.1-mCherry (magenta) and DAPI (grey). *Pcp2*, Purkinje cell protein 2 promoter; cre, recombinase; *Rosa26*, ubiquitous promoter/enhancer; STOP, sequence interrupting gene expression; tdTomato, red fluorescent protein; Kir2.1, inward-rectifier potassium channel; mCherry, red fluorescent protein. Scale bar, 500 μm (**c, f**).

Achieving labeling of all Purkinje cells in the inducible *Pcp2*^{creER};Kir2.1 mice at comparable stages required daily tamoxifen injections, due to the developmental expression pattern of the *Pcp2* gene, which significantly increased neonatal stress and mortality. An overview of all experimental designs is now provided in **Supplementary Fig. 14**. These additions clarify that labeling patterns are consistent across mouse strains and tamoxifen induction protocols, ensuring that the observed phenotypes are not attributable to differences in recombination efficiency. The

corresponding text and images have been incorporated into the **Methods** section (pages 20-21, lines 475-488) and **Supplementary Information** in the revised manuscript.

- Functional assessment of PC activity: it is critical to evaluate Purkinje cell activity in vivo, as well as the output activity of the cerebellum in the deep cerebellar nuclei (DCN).

Response: In response to the reviewer's suggestions, we performed a new set of *in vivo* extracellular recordings to assess Purkinje cell activity and cerebellar output in the cerebellar nuclei. Specifically, we recorded Purkinje cells and cerebellar nuclei neurons at P28 ± 3 days and in adulthood in *Pcp2^{cre/+};Kir2.1* and their control littermates, *Pcp2^{+/+};Kir2.1*. Purkinje cells were identified based on the presence of both simple and complex spikes, as well as the characteristic pause in simple spikes following each complex spike. In juvenile mice, the simple spike firing rate was significantly reduced in *Pcp2^{cre/+};Kir2.1* Purkinje cells compared with controls, and mutant Purkinje cells exhibited increased irregularity in simple spike firing. Conversely, the complex spike rate was significantly higher in *Pcp2^{cre/+};Kir2.1* Purkinje cells than in controls, while its regularity remained unchanged. Simple spike firing primarily reflects the intrinsic activity of Purkinje cells, as previously shown⁴, thus, these findings demonstrate that intrinsic activity is reduced *in vivo* in Kir2.1-overexpressing Purkinje cells. Importantly, despite reduced intrinsic activity, these neurons retained the ability to generate action potentials when stimulated (new **Figure 1f-k**). Recordings from the cerebellar nuclei neurons in *Pcp2^{cre/+};Kir2.1* mice revealed significantly higher firing rates and increased firing irregularity relative to controls (new **Figure 2k-m**). To further investigate the neural correlates of the behavioral deficits we observed in the adult mutants, we also performed *in vivo* recordings in adult mice. Consistent with the juvenile data, adult *Pcp2^{cre/+};Kir2.1* Purkinje cells exhibited significantly lower simple spike firing rates and greater firing irregularity than controls, while complex spike rate and regularity were unchanged. Surprisingly, cerebellar nuclei neurons in adult mutant mice showed reduced firing rates, whereas firing regularity was unaffected (new **Supplementary Fig. 5**). We have incorporated these new data into the revised **Methods** (pages 22-23, lines 534-562), **Results** (pages 5, 7-8, lines 90-101, 120-125, 156-162), and **Discussion** (pages 14, 17, lines 311-315, 391-401) sections.

- Integration of parallel fiber and climbing fiber inputs: understanding how parallel fiber and climbing fiber inputs are integrated by Purkinje cells is essential to interpret the behavioral outcomes. For example, see PMID: 19805323. Additionally, assessing climbing fiber multi-

innervation (e.g., through acute slice experiments) would be informative. At the very least, VGlut2 staining to evaluate climbing fiber inputs is missing.

Response: We thank the reviewer for this important suggestion. While our initial study did not directly assess climbing fiber or parallel fiber innervation, we have now performed additional immunohistochemical analyses using VGLUT2 and VGLUT1 as markers of climbing fiber and parallel fiber terminals, respectively. Alterations in excitatory inputs onto Purkinje cells are a common pathological feature across multiple ataxias, including loss or impaired climbing fibers⁵⁻⁸ and deficits in parallel fiber synapses^{9, 10}. Using VGLUT2 to label climbing fiber terminals, we observed a significant reduction in the percentage of climbing fiber extension within the molecular layer in *Pcp2^{cre/+};Kir2.1* mutants compared with controls, as well as a decrease in VGLUT2 puncta density. In addition, the number of VGLUT2 puncta apposed to the Purkinje cell soma was increased in mutants, indicating defective climbing fiber translocation and impaired synaptic maturation. Analysis of parallel fiber terminals using VGLUT1 revealed a marked reduction in puncta density and an increase in bouton size in mutants relative to controls, consistent with disrupted parallel fiber synaptic organization. Collectively, these results demonstrate that suppressing Purkinje cell intrinsic activity from early development to adulthood disrupts the normal refinement of both climbing fiber and parallel fiber excitatory inputs. This synaptic disorganization underlies the impaired motor coordination, deficits in cerebellum-dependent learning, and altered *in vivo* firing patterns observed in mutant mice (new **Supplementary Fig. 6**). These findings suggest that reduced intrinsic Purkinje cell activity interferes with the maturation of excitatory synapses, ultimately affecting cerebellar circuit function and behavior. We have incorporated these new data and their interpretation into the **Methods** (pages 25-26, lines 613-630), **Results** (page 8, lines 162-172), and **Discussion** (pages 17-18, lines 411-425) sections to highlight and interpret their relevance for excitatory synaptic development and behavioral outcomes.

- PC-DCN immunostaining: immunostaining in non-conditional mice or mice injected with high doses of tamoxifen provides limited information, as co-localization staining lacks functional relevance. Functional data, such as electrophysiological recordings from brain slices, would be far more informative.

Response: We thank the reviewer for their comment and apologize for any confusion regarding the purpose of the Purkinje cell–cerebellar nuclei neuron immunostaining analysis. The primary

objective of this experiment was not to infer functional connectivity, but rather to assess whether Purkinje cell axons appropriately reach and target cerebellar nuclei neurons during development in our control and mutant mice. This experiment was intended to provide an anatomical framework for understanding how reduced intrinsic Purkinje cell activity might alter the structural maturation of cerebellar circuitry. To address the reviewer's concern about functional relevance, we emphasize that cerebellar nuclei neuron activity was directly assessed through *in vivo* electrophysiological recordings in both juvenile and adult mice (new **Figure 2k-m** and **Supplementary Fig. 5g-i**). These recordings provide complementary functional data that, together with the anatomical observations, help to build a mechanistic link between disrupted Purkinje cell intrinsic activity, altered cerebellar output, and the behavioral impairments observed. In the revised manuscript, we have clarified the rationale for the Purkinje cell–cerebellar nuclei neuron immunostaining in the **Results** (pages 6-8, 10, lines 115-125, 160-162, 202-215) and expanded the **Discussion** (pages 15, 17, lines 331-340, 391-401) to explain how the anatomical and functional findings together inform potential mechanisms underlying the deficits in motor coordination and cerebellum-dependent learning.

Furthermore, the switch to high-dose tamoxifen when studying PC-DCN connections during development is abrupt and unexplained. Was the single-dose protocol used in Figure 1 ineffective?

Response: We apologize for the lack of clarity regarding the tamoxifen protocols used in different experiments. As noted earlier, the sparse-labeling strategy was employed specifically to visualize and quantify the dendritic arborization of individual Purkinje cells, where minimal overlap between neighboring cells is essential. This required a low-dose, single-tamoxifen induction to ensure sparse recombination. In contrast, the analysis of Purkinje cell–cerebellar nuclei neuron connectivity and the behavioral experiments required all Purkinje cells to be labeled. For these purposes, we used a high-dose tamoxifen protocol to achieve recombination in all Purkinje cells. Thus, the shift in tamoxifen dosage reflects different experimental goals, sparse labeling for single-cell morphological analyses versus labeling of all Purkinje cells for circuit-level anatomical and behavioral studies, rather than the single-dose protocol shown in **Supplementary Fig. 1d**. To address the reviewer's concern, we have clarified these methodological distinctions in the revised **Methods** section (pages 20-21, lines 475-488), and included representative control images to illustrate the effectiveness of both sparse and complete labeling approaches in their

respective contexts. An overview of all experimental designs is provided in **Supplementary Fig. 14**.

- Non-conditional expression extrapolations: the activity of PCs in $Pcp2^{cre};Kir2.1$ (non-conditional) mice was not tested, yet conclusions are extrapolated from the conditional model injected with tamoxifen at P1.

Response: We appreciate the reviewer's concern regarding the extrapolation from the inducible to the non-inducible conditional models. To directly address this, we have now included *in vivo* Purkinje cell activity recordings from the non-inducible conditional model $Pcp2^{Cre/+};Kir2.1$ and control littermates in the revised manuscript (new **Figure 1f-k**, and **Supplementary Fig. 5a-f**). Importantly, the Kir2.1-mCherry construct is identical in both models. In both cases, Kir2.1 expression is activated only after Cre-mediated excision of the floxed stop cassette. Thus, it is the design of the floxed allele, not the type of Cre driver, that determines expression. Once recombination occurs, whether constitutively or following tamoxifen induction, the excision is permanent, and Kir2.1 is expressed from the same promoter in both systems. Therefore, any potential differences would arise solely from differences in the timing of Cre activity, not from the Kir2.1 system itself (new **Supplementary Fig. 1**). While inducible systems can exhibit variability in recombination efficiency, we carefully validated recombination in our lines. Our immunohistochemical analyses (new **Supplementary Fig. 9** and **Supplementary Fig. 10**) demonstrate consistent labeling of all Purkinje cells in the inducible model. Additionally, prior to initiating this study, we systematically tested our Cre lines to map the timing and pattern of recombination during early development. To improve clarity, we now include **Reviewers Fig. 1**, showing tdTomato and Kir2.1-mCherry expression patterns in $Pcp2^{Cre/+}$ at P1.

Similarly, VOR and eyeblink conditioning experiments were only performed in the non-conditional model. This discrepancy should be addressed.

Response: We thank the reviewer for raising this point. The VOR and eyeblink conditioning experiments were performed in the non-inducible conditional $Pcp2^{cre/+};Kir2.1$ model, and we recognize the importance of explaining why these behavioral assays were not repeated in the inducible line. Because the Kir2.1 construct is identical in both the non-inducible and inducible conditional models, with Kir2.1 expression activated only after Cre-mediated excision of the same

floxed stop cassette (new **Supplementary Fig. 1**), the two systems differ solely in the timing of recombination. As described above, both models show highly consistent anatomical, electrophysiological, and behavioral phenotypes, indicating that Kir2.1 expression has comparable functional consequences regardless of whether recombination occurs constitutively or via tamoxifen induction. Our rationale for using the inducible conditional model was specifically to manipulate the timing of Kir2.1 expression, enabling us to examine how perturbing Purkinje cell intrinsic activity at intermediate developmental stages affects circuit maturation and later behavioral outcomes. To assess these behavioral outcomes, we selected the balance beam test. Because the non-conditional model already provided robust deficits in cerebellum-dependent learning (VOR and eyeblink conditioning), and because the inducible model served a different mechanistic purpose (timing-specific disruption), we did not consider repeating the conditioning experiments essential. We hope this additional explanation clarifies the rationale behind our experimental design.

- A diagram summarizing the experimental design, including the different strains of mice, tamoxifen doses, and the resulting effects, is necessary to improve clarity and guide readers through the manuscript.

Response: We thank the reviewer for this helpful suggestion. To enhance clarity and guide readers through the different experimental strategies, we have now included a schematic diagram summarizing the mouse strains used, the tamoxifen protocols, and the associated experimental outcomes (new **Supplementary Fig. 14**). This visual overview is intended to complement the methodological descriptions and provide a clear framework for interpreting the data across models, developmental stages, and experimental approaches. We hope this addition addresses the reviewer's concern and improves the accessibility of our manuscript.

Reviewer #2 (Remarks to the Author):

This is an elegant and well-written manuscript that tries to demonstrate the role that Purkinje cell activity plays in the development of the cerebellum. Using an inducible transgenic gene approach, the authors explore the temporal properties of intrinsic activity of Purkinje cells in the proper formation of cerebellar circuits.

The figures are clear and the experiments are for the most part quite thorough. The authors try to reduce Purkinje cell activity both early during development and later using inducible expression of Kir2.1 at different ages. Thus, the authors try to establish that its role is most important during early development, with the strongest effects on Purkinje cell dendritic length and behavior being observed when activity is impaired at the earliest postnatal ages.

Overall I think that this manuscript reasonably demonstrates the importance of Purkinje cell activity in Purkinje cell dendritic growth and in cerebellar function, although more evidence about exactly what is happening with firing would help the manuscript (see below). If these changes are incorporated, I believe that it will be a classic paper that will be well cited both in the cerebellar field and in the field of circuit development, meaning its impact will be large and thus is suitable for publication in Nature Comms.

While I think that the manuscript is very interesting, I am not yet fully convince by the data that expressing Kir2.1 is doing so via Purkinje cell activity and not some other means (such as making the cells sick). Points 1 and 2 would clarify this and provide additional strength to their hypothesis.

Response: We thank the reviewer for their positive and constructive evaluation of our manuscript. We particularly appreciate the recognition of the potential impact of our findings for both the cerebellar and broader circuit development fields.

1. The firing data presented in Extended Data Figure 1d-e shows that the Purkinje cells expressing Kir2.1 at 21 days old can still spike, although it could be highlighted better by showing current injection steps for additional representative traces to better demonstrate the right-ward shift, which is unfortunately not particular evidence with the 3 traces shown for each example. It would be beneficial to see traces with fewer action potentials so a better comparison of individual action potentials could be observed as well.

Response: We thank the reviewer for this helpful suggestion. To better illustrate the reduced excitability of Kir2.1-expressing Purkinje cells at P21, we have expanded the set of representative traces to include a broader range of current injection steps (new **Supplementary Fig. 2f**). These additional examples more clearly reveal the rightward shift in the firing–current relationship and the reduced spike output, complementing the data in **Figure 1d-e**. We have also included traces from lower-current injection steps that evoke fewer action potentials, enabling improved visualization and comparison of individual spikes between control and Kir2.1-expressing Purkinje cells.

I am surprised that there is no activity at 0 pA for the two control conditions given that Purkinje cells typically fire spontaneously at P21, and would like the authors to comment on this.

Response: We thank the reviewer for raising this important point. We apologize for the lack of clarity. The "0 pA" condition in our current-step protocol does not correspond to the cells' natural resting membrane potential. In this experiment, a holding current is used to clamp each Purkinje cell at -65 mV. This holding current is then labeled as "0 pA" prior to applying current-injection steps. We have now clarified this point in the **Methods** (page 22, lines 513-528) to avoid confusion.

Also, the magnitude of the voltage deflection to the negative current injection is much smaller in the Kir2.1 expressing cells, likely because the cells are smaller. I think this could be supported by reporting the whole cell capacitance and input resistance (R_{in}) for their recordings. I think this data should be incorporated into Fig. 1 as without that, the impression arising from Figure 1 is

that firing is completely abolished, which is likely incorrect (and also brings up the question of whether the cells can fire action potentials at all).

Response: We thank the reviewer for raising this important point. To clarify whether the effects of Kir2.1 expression reflect suppression of Purkinje cell activity rather than general cellular dysfunction, we have now quantified and included input resistance (R_{in}) from our recordings. As expected, Kir2.1-overexpressing cells show a significantly reduced R_{in} , consistent with increased potassium conductance (new **Supplementary Fig. 2e**). Regarding whole-cell capacitance, accurate measurement in Purkinje cells is challenging due to their large and intricate architecture, which violates the assumption of membrane isopotentiality¹¹. Additionally, intracellular calcium concentrations can influence membrane capacitance^{12, 13}, and Kir2.1-overexpressing Purkinje cells exhibit altered calcium signaling. Despite these limitations, we plotted capacitance values alongside rheobase in **Reviewers Fig. 2** and found no significant differences in capacitance between control ($Pcp2^{creER};Ai14$, $Pcp2^{creER};Kir2.1,Ctl$) and mutant ($Pcp2^{creER};Kir2.1$) Purkinje cells at P21. Importantly, the Purkinje cells exhibited a higher rheobase, indicating that more current is required to elicit action potentials.

Reviewers Fig. 2 Overexpression of Kir2.1 increases the rheobase of Purkinje cells at P21.

a Quantification of the capacitance from $Pcp2^{creER};Ai14$ ($n = 11$ cells/3 mice), $Pcp2^{creER};Kir2.1,Ctl$ ($n = 11$ cells/4 mice) and $Pcp2^{creER};Kir2.1$ ($n = 11$ cells/4 mice), Kruskal-Wallis multiple comparisons test. **b** Quantification of Rheobase in Purkinje cells from $Pcp2^{creER};Ai14$ ($n = 11$ cells/3 mice), $Pcp2^{creER};Kir2.1,Ctl$ ($n = 11$ cells/4 mice) and $Pcp2^{creER};Kir2.1$ ($n = 13$ cells/4 mice). One-way ANOVA Bonferroni's multiple comparisons test: *** $P < 0.001$. Data related to Fig. 1.

After achieving whole-cell configuration, zero current was injected in current-clamp mode to assess intrinsic action potential firing and resting membrane potential, defined as the mode of the voltage trace. In control recordings, 77% of Purkinje cells exhibited spontaneous firing, confirming intrinsic activity. In contrast, Kir2.1 overexpression abolished spontaneous firing in 90% of the recorded $Pcp2^{creER};Kir2.1$ Purkinje cells by hyperpolarizing the resting membrane potential (new

Figure 1a-c). However, when we applied current steps in this configuration, we found that every *Pcp2^{creER};Kir2.1* Purkinje cell was able to fire action potentials if the current steps were large enough (new **Figure 1e** and **Supplementary Fig. 2f**). These data clarify that Purkinje cells expressing Kir2.1 can still fire action potentials when depolarized, but spontaneous activity is largely abolished. We have incorporated these new data and their interpretation into the **Methods** (page 22, lines 513-528) and **Results** (page 5, lines 75-89) sections.

Additionally, we performed a new set of *in vivo* extracellular recordings to assess Purkinje cell activity and cerebellar output in the cerebellar nuclei. Specifically, we recorded Purkinje cells and cerebellar nuclei neurons at P28 ± 3 days and in adulthood in *Pcp2^{cre/+};Kir2.1* and their control littermates, *Pcp2^{+/+};Kir2.1*. In juvenile mice, the simple spike firing rate was significantly reduced in *Pcp2^{cre/+};Kir2.1* Purkinje cells compared with controls, and mutant Purkinje cells exhibited increased irregularity in simple spike firing. Conversely, the complex spike rate was significantly higher in *Pcp2^{cre/+};Kir2.1* Purkinje cells than in controls, while its regularity remained unchanged. As previously shown, simple spike firing primarily reflects the intrinsic activity of Purkinje cells⁴, thus, these findings demonstrate that intrinsic activity is reduced *in vivo* in Kir2.1-overexpressing Purkinje cells. Importantly, despite reduced intrinsic activity, these neurons retained the ability to generate action potentials when stimulated (new **Figure 1f-k**). Recordings from the cerebellar nuclei neurons in *Pcp2^{cre/+};Kir2.1* mice revealed significantly higher firing rates and increased firing irregularity relative to controls (new **Figure 2k-m**). To investigate the neural correlates of the behavioral deficits we observed in the adult mutants, we also performed *in vivo* recordings in adult mice. Consistent with the juvenile data, adult *Pcp2^{cre/+};Kir2.1* Purkinje cells exhibited significantly lower simple spike firing rates and greater firing irregularity than controls, while complex spike rate and regularity were unchanged. Cerebellar nuclei neurons in adult mutant mice showed reduced firing rates, whereas firing regularity was unaffected (new **Supplementary Fig. 5**). We have incorporated these new data into the revised **Methods** (pages 22-23, lines 534-562), **Results** (pages 5-8, lines 90-101, 120-125, 156-162), and **Discussion** (pages 14, 17, lines 311-315, 391-401) sections.

2. How Kir2.1 affects Purkinje cell activity at young ages is demonstrated in Extended Figure 4. However, the authors do not focus on activity, but rather report membrane potential and the amplitude of a current evoked at a negative holding potential. This seems counter-intuitive to me, given that we know that Purkinje cells fire action potentials at these young ages. Firing data would

be much more convincing and informative in my opinion, since firing is what they are trying to manipulate. I suggest that this is done as described above in point 2. The authors argue that the effects of Kir2.1 are mediated through the alteration in firing activity, so I think that needs to be shown directly. Without direct evidence of firing differences, it could be that Kir2.1 is having an effect by making the cells sicker or decreasing their Rin and making them leakier rather than directly through firing.

Response: To address the reviewer's concern, we quantified and included input resistance (Rin) values from our recordings. Kir2.1-overexpressing cells do indeed show a significantly reduced Rin, consistent with the increased potassium conductance introduced by Kir2.1 (new **Supplementary Fig. 8d, h, l**). However, *ex vivo* recordings demonstrate that all cells can still fire action potentials. *In vivo* recordings confirm that this is not just a hypothetical scenario, but, in fact, all Purkinje cells show simple spike activity, ranging from 111 to 0.06 Hz (new **Figure 1h**). Thus, the effect of synaptic input on the cell could be altered, but the main effect is a reduction in membrane potential, leading to reduced intrinsic firing activity. Although morphological development is impaired (new **Figure 2a-g**), Calbindin staining revealed no evidence for neurodegeneration at any age, including in the adult mice used for behavioral testing. As noted above, capacitance measurements are complex to interpret in developing Purkinje cells due to non-isopotentiality and ongoing morphological changes. However, we plotted the capacitance values in **Reviewers Fig. 3** and found a significant reduction in capacitance between control and mutant Purkinje cells at P7, P14 or P21.

To directly assess firing activity, we performed *ex vivo* recordings at each developmental stage. After achieving whole-cell configuration, no current was injected in current-clamp mode in order to measure intrinsic action potential firing and resting membrane potential. In control recordings, 77%, 92%, and 81% of Purkinje cells exhibited spontaneous firing at P7, P14, and P21, respectively. In contrast, only 6%, 0% and 8% of *Pcp2^{creER};Kir2.1* Purkinje cells were intrinsically active at the corresponding ages (new **Supplementary Fig. 8a, e, i**). These data directly demonstrate that Kir2.1 effectively suppresses intrinsic firing throughout development and support our conclusion that the observed cellular and circuit effects arise from reduced intrinsic activity rather than nonspecific cellular dysfunction. We have incorporated these new data and

their interpretation into the **Methods** (page 22, lines 513-528) and **Results** (pages 9-10, lines 195-202) sections.

Reviewers Fig. 3 Overexpression of Kir2.1 decreases the capacitance of Purkinje cells at different developmental ages. **a** Quantification of capacitance in control (n = 12 cells/3 mice) and *Pcp2^{creER};*Kir2.1 (n = 12 cells/4 mice) Purkinje cells at P7. Mann-Whitney *U* test: ****P* < 0.001. **b** Quantification of capacitance in control (n = 17 cells/2 mice) and *Pcp2^{creER};*Kir2.1 (n = 15 cells/5 mice) Purkinje cells at P14. Unpaired Student's *t*-test: ****P* < 0.001. **c** Quantification of capacitance in control (n = 26 cells/3 mice) and *Pcp2^{creER};*Kir2.1 (n = 12 cells/5 mice) Purkinje cells at P21. Mann-Whitney *U* test: ****P* < 0.001. Data related to Fig. 4.

3. The manuscript focuses heavily on Purkinje cell structure, and relate this to behavior. Yet Purkinje cells are known to influence the development of other parts of the cerebellar circuit. It would be useful if the authors could determine if other parts of the circuit are changed by this manipulation of Purkinje cell activity. While I do not think it is required to characterize everything, at least the size of the lobules and the thickness of the granule cell layer (as a read-out of granule cell number), would be informative. The MLI number would also be interesting, since there are dendritic changes in Purkinje cells. This would help understand the extent of the changes in the circuit and whether behavioral changes are likely to arise entirely from Purkinje-cell related changes or may also involve other parts of the circuit. I think this is most important for the P1 and P7 injections since this is when granule cells and MLIs are still migrating.

Response: We appreciate the reviewer's suggestion to examine whether manipulating Purkinje cell activity alters additional components of the cerebellar circuit. To address this, we have now

analysed cerebellar lobule size, granule cell layer and molecular layer thickness, and molecular layer interneuron number in adult mice subjected to Kir2.1-overexpression from either P1 or P7 through adulthood (new **Supplementary Fig. 11**). Because delaying the onset of Kir2.1-overexpression by just one week was sufficient to improve balance beam performance, we hypothesized that the early deficits observed in the most severe phenotype, where Kir2.1 was overexpressed from birth, might also impair the maturation of other cerebellar cell types whose development depends on proper Purkinje cell maturation, namely molecular layer interneurons (MLIs)^{14, 15} and granule cells (GCs)^{16, 17}. To test this, we examined MLI density in adult *Pcp2^{cre/+};Kir2.1* mice and their respective controls. While parvalbumin (PV) is a known marker for both Purkinje cells and MLIs¹⁸, it does not label all MLIs within the molecular layer¹⁹. Hence, we quantified the number of PV-positive (PV+) cells co-labeled with NeuroTrace (NeuT+), a fluorescent Nissl stain, within the molecular layer. Interestingly, the number of PV+ NeuT+ cells, but not NeuT+ cells, was reduced in *Pcp2^{cre/+};Kir2.1* mice compared with controls. The molecular layer was also thinner than in controls. We next assessed MLI density in mice in which Purkinje cell activity was disrupted at P7, thereby sparing the first week of development from changes in activity. The density of PV+ NeuT+ and NeuT+ MLIs, as well as the molecular layer thickness, were unchanged between groups. Lastly, we analysed granule cell layer thickness (GCL) as a proxy for GC numbers and measured the size of vermal cerebellar lobules in *Pcp2^{cre/+};Kir2.1* mice and in P7-inducible *Pcp2^{creER};Kir2.1* mice and their respective controls. There was no difference in the GCL thickness of the cerebellar cortex between *Pcp2^{cre/+};Kir2.1* or P7 inducible *Pcp2^{creER};Kir2.1* mice and controls. However, lobules I-II, IV-V, and IX were smaller in *Pcp2^{cre/+};Kir2.1* mice, while lobules IV-V and VI were reduced in the P7-inducible *Pcp2^{creER};Kir2.1* mice compared to controls. Together, these results indicate that early disruption of Purkinje cell intrinsic activity not only leads to more severe motor deficits but also more strongly interferes with the proper maturation of cerebellar circuitry and structure. These new data help to further characterize the extent of circuit-wide changes induced by manipulating Purkinje cell activity and provide important context for interpreting the behavioral changes observed in our experiments. We have incorporated these analyses into the **Methods** (pages 25-26, lines 602-605, 631-641), **Results** (page 11, lines 229-251), and **Discussion** (page 16, lines 361-368) sections to include this additional analysis.

4. Finally, the authors state that they label only a subset of Purkinje cells with these methods. It would be beneficial to understand approximately how many Purkinje cells are labeled and in which

lobules they tend to see labeling, if indeed there is some regionalization of expression of their Kir2.1 transgene.

Response: We apologize for the lack of clarity regarding our experimental design in the original manuscript. The goal of the experiments shown in new **Figure 2a-g** and **Figure 4** was to investigate whether disrupting intrinsic activity impacted Purkinje cell morphological development. To enable high-resolution dendritic reconstruction in a densely packed brain structure, we deliberately employed a sparse labeling strategy using the *Pcp2^{creER}* line with a single tamoxifen injection (100mg/kg) at postnatal day 1. This approach generated sparse tdTomato or Kir2.1-mCherry expression in Purkinje cells, preventing overlap with neighboring cells and thereby allowing reliable quantification of individual dendritic arbors. To further validate the robustness of our findings, we obtained comparable results using a different approach for sparse labeling, i.e., using *in utero* electroporation, as shown in the new **Supplementary Fig. 3**. Representative images of sparse labeling are now included in the revised manuscript in **Supplementary Fig. 1a-d** and **Supplementary Fig. 7**. While we have not quantified the precise number of labeled Purkinje cells, the representative images demonstrate broad and evenly distributed expression throughout the cerebellar cortex. We did not observe any systematic regional bias, although some lobular variability in expression intensity is apparent, which is typical for *Pcp2*-driven expression. For all other experiments, *in vivo* extracellular recordings, synaptic terminal quantification, behavioral tests, and RNA sequencing, we employed a genetic strategy to achieve labeling of all Purkinje cells. This was accomplished either by modifying tamoxifen dosing and injection frequency in the inducible *Pcp2^{creER};Ai14* and *Pcp2^{creER};Kir2.1* mice, or by using the non-inducible *Pcp2^{cre/+};Kir2.1* mouse model. Corresponding images of these protocols have been added to **Supplementary Fig. 1e-g**, **Supplementary Fig. 9**, and **Supplementary Fig. 10**.

For experiments requiring recombination from birth, while avoiding multiple injections in neonatal pups, we used non-inducible *Pcp2^{cre/+};Kir2.1* mice and their littermate controls. This ensured Kir2.1-mCherry expression in all Purkinje cells from the earliest developmental stages. Previous studies detected *Pcp2* mRNA at P1¹ and GFP expression at E17 in a *Pcp2-GFP* transgenic mice². We found that in *Pcp2^{cre/+}(B6.Cg-Tg(Pcp2-cre)3555Jdhu/J)*³ we could detect tdTomato or Kir2.1-mCherry at P1 in *Pcp2^{cre/+};Ai14* and *Pcp2^{cre/+};Kir2.1* mice, respectively (**Reviewers Fig. 1** below). At P1 it is possible to distinguish the parasagittal bands of tdTomato (**Reviewers Fig. 1c**) and Kir2.1-mCherry (**Reviewers Fig. 1c**) expression characteristic of the *Pcp2* gene at this age. By P7, *Pcp2^{cre/+};Kir2.1* mice show labeling in all Purkinje cells, but with a higher intensity in the nodular regions of the cerebellum due to the *Pcp2* gene developmental expression pattern (this

gene is first expressed in parasagittal bands overlapping Zebrin-positive cells, and then the expression extends to all Purkinje cells) (new **Supplementary Fig. 9a, b**). Achieving labelling of all Purkinje cells in the inducible *Pcp2^{creER};Kir2.1* mice at comparable stages required daily tamoxifen injections due to the expression pattern of *Pcp2* gene, which significantly increased neonatal stress and mortality. An overview of all experimental designs is now provided in **Supplementary Fig. 14**. These additions clarify that labeling patterns are consistent across mouse strains and tamoxifen induction protocols, ensuring that the observed phenotypes are not attributable to differences in recombination efficiency. The corresponding text and images have been incorporated into the **Methods** section (pages 20-21, lines 475-488) and **Supplementary Information** in the revised manuscript.

Reviewers Fig. 1 Specific labeling of Purkinje cells at P1 in *Pcp2^{cre/+};Ai14* and *Pcp2^{cre/+};Kir2.1*

a Experimental design. Brain from conditional *Pcp2^{cre/+};Ai14* was collected at P1. **b** Schematic illustrating the genetic strategy used to express tdTomato specifically in Purkinje cells, showing that upon Cre-mediated recombination, it is an all-or-none event. **c** Coronal cerebellar sections at P1 from *Pcp2^{cre/+};Ai14* mice show labeling of tdTomato (magenta) and DAPI (grey). **d** Experimental design. Brain from conditional *Pcp2^{cre/+}; Kir2.1* mice was collected at P1. **e** Schematic illustrating the genetic strategy used to express Kir2.1-mCherry specifically in Purkinje cells. **f** Coronal cerebellar sections at P1 from *Pcp2^{cre/+};Kir2.1* mice show labeling of Kir2.1-mCherry (magenta) and DAPI (grey). *Pcp2*, Purkinje cell protein 2 promoter; cre, recombinase; *Rosa26*, ubiquitous promoter/enhancer; STOP, sequence interrupting gene expression;

tdTomato, red fluorescent protein; Kir2.1, inward-rectifier potassium channel; mCherry, red fluorescent protein. Scale bar, 500 μm (c, f).

References

1. Oberdick, J., Levinthal, F. & Levinthal, C. A Purkinje cell differentiation marker shows a partial DNA sequence homology to the cellular sis/PDGF2 gene. *Neuron* **1**, 367-376 (1988).
2. Tomomura, M., Rice, D.S., Morgan, J.I. & Yuzaki, M. Purification of Purkinje cells by fluorescence-activated cell sorting from transgenic mice that express green fluorescent protein. *Eur J Neurosci* **14**, 57-63 (2001).
3. Zhang, X.M., *et al.* Highly restricted expression of Cre recombinase in cerebellar Purkinje cells. *Genesis* **40**, 45-51 (2004).
4. Zhou, H., *et al.* Cerebellar modules operate at different frequencies. *Elife* **3**, e02536 (2014).
5. Osorio, C., *et al.* Pre-ataxic loss of intrinsic plasticity and motor learning in a mouse model of SCA1. *Brain* **146**, 2332-2345 (2023).
6. Ebner, B.A., *et al.* Purkinje cell ataxin-1 modulates climbing fiber synaptic input in developing and adult mouse cerebellum. *J Neurosci* **33**, 5806-5820 (2013).
7. Furrer, S.A., *et al.* Reduction of mutant ataxin-7 expression restores motor function and prevents cerebellar synaptic reorganization in a conditional mouse model of SCA7. *Hum Mol Genet* **22**, 890-903 (2013).
8. Lin, C.C., *et al.* Reduced cerebellar rhythm by climbing fiber denervation is linked to motor rhythm deficits in mice and ataxia severity in patients. *Sci Transl Med* **17**, eadk3922 (2025).
9. Tempia, F., *et al.* Parallel fiber to Purkinje cell synaptic impairment in a mouse model of spinocerebellar ataxia type 27. *Front Cell Neurosci* **9**, 205 (2015).
10. Kurihara, H., *et al.* Impaired parallel fiberPurkinje cell synapse stabilization during cerebellar development of mutant mice lacking the glutamate receptor delta2 subunit. *J Neurosci* **17**, 9613-9623 (1997).
11. Golowasch, J., *et al.* Membrane capacitance measurements revisited: dependence of capacitance value on measurement method in nonisopotential neurons. *J Neurophysiol* **102**, 2161-2175 (2009).
12. Balantic, K., Weiss, V.U., Allmaier, G. & Kramar, P. Calcium ion effect on phospholipid bilayers as cell membrane analogues. *Bioelectrochemistry* **143**, 107988 (2022).
13. Carter, T.D., Zupancic, G., Smith, S.M., Wheeler-Jones, C. & Ogden, D. Membrane capacitance changes induced by thrombin and calcium in single endothelial cells cultured from human umbilical vein. *J Physiol* **513 (Pt 3)**, 845-855 (1998).
14. Telley, L., *et al.* Dual Function of NRP1 in Axon Guidance and Subcellular Target Recognition in Cerebellum. *Neuron* **91**, 1276-1291 (2016).
15. Fleming, J.T., *et al.* The Purkinje neuron acts as a central regulator of spatially and functionally distinct cerebellar precursors. *Dev Cell* **27**, 278-292 (2013).
16. Wechsler-Reya, R.J. & Scott, M.P. Control of neuronal precursor proliferation in the cerebellum by Sonic Hedgehog. *Neuron* **22**, 103-114 (1999).
17. Lewis, P.M., Gritli-Linde, A., Smeyne, R., Kottmann, A. & McMahon, A.P. Sonic hedgehog signaling is required for expansion of granule neuron precursors and patterning of the mouse cerebellum. *Dev Biol* **270**, 393-410 (2004).
18. Stichel, C.C., Kagi, U. & Heizmann, C.W. Parvalbumin in cat brain: isolation, characterization, and localization. *J Neurochem* **47**, 46-53 (1986).
19. Schilling, K. & Oberdick, J. The treasury of the commons: making use of public gene expression resources to better characterize the molecular diversity of inhibitory interneurons in the cerebellar cortex. *Cerebellum* **8**, 477-489 (2009).

Response to reviewers for Nature Communications manuscript NCOMMS-24-63742A by Osório et al.

We sincerely thank the reviewers for their insightful and constructive comments. Their feedback has been invaluable in improving the clarity, rigor, and overall quality of our manuscript.

Reviewer #1 (Remarks to the Author):

The authors have convincingly addressed my main concerns. They performed additional experiments and provided extensive analyses that clarify ambiguities present in the previous version of the manuscript. The revised manuscript is also easier to read and more clearly demonstrates the importance of Purkinje cell intrinsic excitability in cerebellar development. Overall, this work represents one of the most comprehensive datasets on this topic compared with prior studies.

*Nonetheless, the following related study should also be cited: Lorenzetto E, Caselli L, Feng G, Yuan W, Nerbonne JM, Sanes JR, Buffelli M. *Genetic perturbation of postsynaptic activity regulates synapse elimination in developing cerebellum.* **Proc Natl Acad Sci U S A.** 2009 Sep 22;106(38):16475–80.*

My remaining concern is that the main Figure 5 should be extended to include the data currently shown in Supplementary Figure 10 (panels I–L). In its current form, Figure 5 does not clearly convey that phenotype severity depends on the timing of induction. Adding these panels would strengthen the figure and support this key conclusion.

Response: We thank the reviewer for their thoughtful and positive evaluation of our manuscript and for recognizing the breadth of the dataset and its contribution to the field. We are pleased that the additional experiments and analyses have clarified the remaining ambiguities and improved the clarity and impact of the manuscript. As suggested, we have now cited the relevant study by Lorenzetto *et al.* (PNAS, 2009), which provides important context regarding the role of postsynaptic activity in cerebellar synapse elimination. This reference has been added to the revised manuscript in the **Discussion** section (pages 17-18, lines 413-419). In response to the reviewer's comment regarding **Figure 5**, we have extended the main figure to include the data previously presented in **Supplementary Fig. 10** (panels i–l). Incorporating these panels into

Figure 5 now more clearly demonstrates that phenotype severity depends on the timing of induction, thereby strengthening the figure and directly supporting this key conclusion. Please find these new changes in the revised manuscript in the **Results** section (pages 10-11, lines 216-228, 1277-1291). We hope the reviewer finds that these changes provide more clarity to the results.

Reviewer #2 (Remarks to the Author):

I am happy with the response to my questions, and feel that the changes incorporated in the revised manuscript significantly strengthen it. I have no further concerns with the manuscript. I think it will add to our understanding of cerebellar development, an important topic.

Response: We thank the reviewer for their positive and constructive evaluation of our manuscript. We are pleased that the revisions have addressed their concerns and strengthened the study. We particularly appreciate their recognition of the contribution of our findings to advancing understanding of cerebellar development and circuit maturation.